# Silencing long ascending propriospinal neurons after spinal cord injury improves hindlimb stepping in the adult rat

Courtney T Shepard[1,2,3], Amanda M Pocratsky[2,3], Brandon L Brown[1,2,3], Morgan A Van Rijswijck[3,4], Rachel M Zalla[3,4], Darlene A Burke[3,5], Johnny R Morehouse[3,5], Amberley S Riegler[3,5], Scott R Whittemore[1,2,3,5], David SK Magnuson[1,2,3,4,5]*

[1]Interdisciplinary Program in Translational Neuroscience, School of Interdisciplinary and Graduate Studies, University of Louisville, Louisville, United States; [2]Department of Anatomical Sciences and Neurobiology, University of Louisville, Louisville, Louisville, United States; [3]Kentucky Spinal Cord Injury Research Center, University of Louisville, Louisville, United States; [4]Speed School of Engineering, University of Louisville, Louisville, United States; [5]Department of Neurological Surgery, University of Louisville, Louisville, United States

*For correspondence:
dsmagn01@louisville.edu

**Abstract** Long ascending propriospinal neurons (LAPNs) are a subpopulation of spinal cord interneurons that directly connect the lumbar and cervical enlargements. Previously we showed, in uninjured animals, that conditionally silencing LAPNs disrupted left-right coordination of the hindlimbs and forelimbs in a context-dependent manner, demonstrating that LAPNs secure alternation of the fore- and hindlimb pairs during overground stepping. Given the ventrolateral location of LAPN axons in the spinal cord white matter, many likely remain intact following incomplete, contusive, thoracic spinal cord injury (SCI), suggesting a potential role in the recovery of stepping. Thus, we hypothesized that silencing LAPNs after SCI would disrupt recovered locomotion. Instead, we found that silencing spared LAPNs post-SCI improved locomotor function, including paw placement order and timing, and a decrease in the number of dorsal steps. Silencing also restored left-right hindlimb coordination and normalized spatiotemporal features of gait such as stance and swing time. However, hindlimb-forelimb coordination was not restored. These data indicate that the temporal information carried between the spinal enlargements by the spared LAPNs post-SCI is detrimental to recovered hindlimb locomotor function. These findings are an illustration of a post-SCI neuroanatomical-functional paradox and have implications for the development of neuronal- and axonal-protective therapeutic strategies and the clinical study/implementation of neuromodulation strategies.

## Editor's evaluation

The authors used viral transduction of tetanus toxin to silence synaptic transmission of long ascending propriospinal neurons (LAPN) to understand the role they play in function following contusion injury to the spinal cord. Kinematic analysis revealed that silencing LAPN improved inter-limb and intralimb coordination rather than worsening it, as expected. This unexpected finding demonstrates that roles of LAPN and other interneurons following spinal cord contusion should not be assumed to mirror function of these neurons in uninjured spinal circuitry.

## Introduction

In mammals, locomotion involves descending commands from supraspinal centers and peripheral input from sensory systems converging on spinal locomotor circuitry. This circuitry includes central pattern generators (CPGs), intrinsic spinal networks that generate the coordinated muscle activity associated with stereotypic limb movements during stepping (*Brown, 1997*; *Sherrington, 1910*; *Sherrington and Laslett, 1903*). Each limb has its own CPG and the lumbar and cervical CPG pairs are interconnected by long ascending (LAPNs) and long descending (LDPNs) propriospinal neurons (*English et al., 1985*; *Giovanelli Barilari and Kuypers, 1969*). These neurons are thought to provide a functional coupling of the two enlargements and presumably allow precise temporal information to be passed between the hindlimb and forelimb CPGs (*English et al., 1985*; *Juvin et al., 2005*; *Miller et al., 1975*; *Rossignol et al., 1993*). However, direct interactions between the two populations have yet to be demonstrated. LAPN somata reside in the intermediate gray matter, primarily in laminae VII and VIII, with 40–60% having commissural axons (*Brockett et al., 2013*; *Reed et al., 2006*; *Reed et al., 2009*) that cross at-level before ascending in the outermost regions of the lateral funiculus and ventrolateral funiculus (VLF) (*Basso et al., 2002*; *Reed et al., 2006*; *Reed et al., 2009*).

Spinal cord injury (SCI) disrupts communication between the brain and spinal cord, resulting in an immediate inability to initiate and maintain patterned weight-supported locomotion at or below the level of lesion (*Côté et al., 2017*; *Dietz and Harkema, 2004*; *Fong et al., 2009*). Even if classified as neurologically complete, most SCIs are anatomically incomplete with some sparing of white matter, most often the outermost rim of the lateral funiculus and VLF where LAPN axons reside (*Brockett et al., 2013*). These neurons and their axons comprise a percentage of the anatomically spared circuitry, thus providing a functional bridge across the injury site, making them well suited to participate in locomotor recovery after incomplete, contusive thoracic SCI (*Conta and Stelzner, 2004*; *Conta Steencken et al., 2011*; *Conta Steencken and Stelzner, 2010*; *Siebert et al., 2010*). Many studies have reinforced this notion, suggesting that propriospinal neurons, albeit descending in these studies, can contribute to locomotor recovery by serving as injury-crossing bridges (*Bareyre et al., 2004*; *Benthall et al., 2017*; *Filli et al., 2014*; *Flynn et al., 2011*; *Vavrek et al., 2006*).

Conditionally silencing LAPNs in uninjured rat resulted in the partial decoupling of the forelimb and hindlimb pairs, disrupting alternation at each girdle when overground locomotion was evaluated on a high friction surface (*Pocratsky et al., 2020*). These changes were independent of locomotor rhythm and speed and did not affect intralimb joint movements or coordination. Given the anatomical characteristics and functional importance of LAPNs for overground locomotion on high friction surfaces, we evaluated the functional contributions of LAPNs post-SCI in the same context. We hypothesized that LAPNs would contribute to recovered function after incomplete thoracic SCI and that silencing them would further disrupt locomotion by reducing the number of functional spared axons that cross the injury site leaving animals unable to produce a functional stepping pattern.

## Results

### Silencing alters interlimb coordination while other key features of locomotion are maintained

To silence LAPNs, we utilized the two-virus strategy developed by Tadashi Isa and colleagues (*Kinoshita et al., 2012*). A highly efficient retrograde lentiviral vector (HiRet) containing a tetracycline response element upstream of enhanced tetanus neurotoxin (eTeNt) and EGFP (eTeNT.EGFP) was injected at C5/6 (LAPN axon terminals) and an AAV2/2 containing a tetracycline transactivator (rtTAV16) was injected at L2/3 (LAPN cell bodies). Doxycycline (Dox) administration produces eTeNT. EGFP in LAPNs, which cleaves the vesicular docking protein VAMP2, preventing neurotransmitter release and silencing LAPNs (*Figure 1a and b*, see Materials and methods for details).

For overground kinematic analyses (*Figure 1c*), we marked the skin overlying the iliac crest, hip, ankle, and toe (*Figure 1d*) to assess intralimb coordination of the proximal (hip/knee) and distal (knee/ankle) limb segment angles. In uninjured animals, we observed normal rhythmic excursions of the proximal and distal limb segments before (control) and during (Dox) silencing of the LAPNs (*Figure 1—figure supplement 1a,e*; *Figure 1—figure supplement 1—source data 1*), as well as coordinated flexor-extensor movements of the proximal and distal angles during normal walking (*Figure 1—figure supplement 1e-g*; *Figure 1—figure supplement 1—source data 1*). LAPN silencing resulted in a

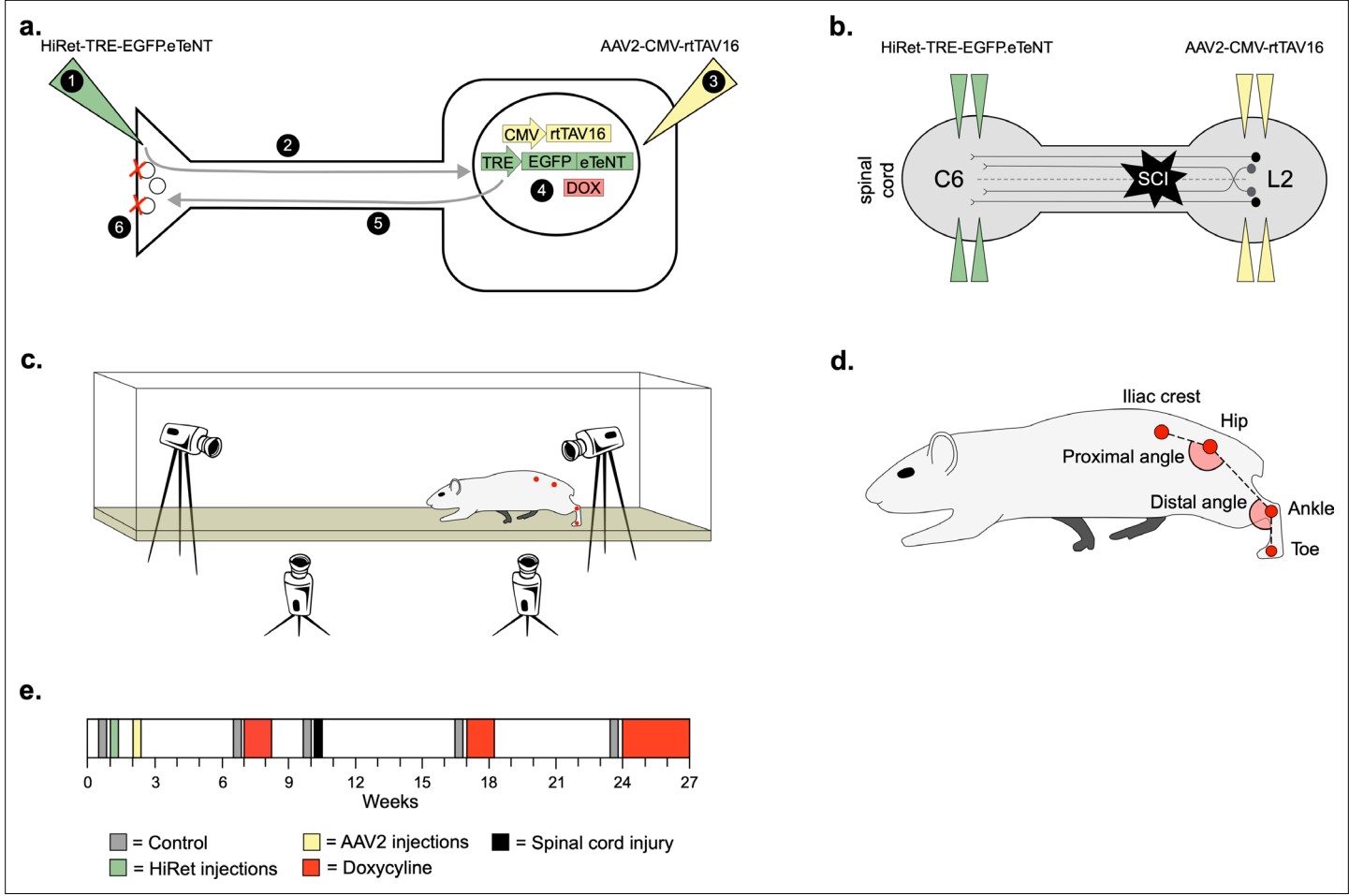

**Figure 1.** Experimental design to conditionally silence long ascending propriospinal neurons (LAPNs) after spinal cord injury (SCI). (**a**) Viral injection protocol for viral silencing in neurons. HiRet-TRE-EGFP.eTeNT ('eTeNT'), a lentiviral vector is injected at axon terminals (**a1**) and is retrogradely transported to cell bodies with high efficiency (**a2**). A second viral vector, AAV2-CMV-rtTAV16, is injected at the cell bodies and expresses the doxycycline (Dox)-dependent Tet-On sequence, a variant of the reverse tetracycline transactivator (**a3**). Without Dox, neurons constitutively express rtTAV sequence, but it is not active. Upon administration of Dox in the animals' drinking water (**a4**), rtTAV16 becomes active and binds to the tetracycline response element (TRE) promoter on the lentiviral vector, leading to expression of EGFP.eTeNT only in doubly infected neurons. EGFP.eTeNT is then anterogradely transported to neuronal cell terminals (**a5**), where it cleaves vesicle associated membrane protein 2 (VAMP2) which impairs vesicles binding to the cell membrane, vesicle exocytosis, and neurotransmission (**a6**, *Pocratsky et al., 2017*; *Kinoshita et al., 2012*). Bilateral injections of enhanced tetanus neurotoxin (eTeNT, green triangles) (**b**) and Tet-On rtTAV16 (yellow triangles, **b**) were performed at C6 and L2 spinal cord levels, respectively, followed by an spinal cord injury (SCI) (black star, **b**). Kinematic measurements were obtained in a clear acrylic tank with two sagittal cameras and one ventral camera to obtain intralimb and interlimb coordination, respectively (**c**). Joint angles were obtained by marking animals' iliac crest, hip, ankle, and toe using tattooing ink and physical palpation of the joints (**d**). The experimental timeline is shown in (**e**).

The online version of this article includes the following figure supplement(s) for figure 1:

**Figure supplement 1.** Long ascending propriospinal neurons (LAPNs) do not contribute to intralimb locomotor coordination of uninjured animals.

**Figure supplement 1—source data 1.** Long ascending propriospinal neurons (LAPNs) do not contribute to intralimb locomotor coordination of uninjured animals.

slight, but statistically significant, increase in the excursion of the distal joint angle during stepping (*Figure 1—figure supplement 1j*; *Figure 1—figure supplement 1—source data 1*), but had no other impact on intralimb coordination.

We examined the coupling patterns of various limb pairs by dividing the initial contact time of one limb by the stride time of the second limb, expressed as a phase value (*Figure 2a*). Phase values of 0 or 1 indicate synchrony (lead-limb dependent) and values of 0.5 indicate alternation, typically shown on circular plots as 0 or 1 at 0° and 0.5 at 180°. For slower gaits such as walk and trot, phase values of hindlimb, forelimb, and homolateral hindlimb-forelimb pairs would be concentrated around 0.5,

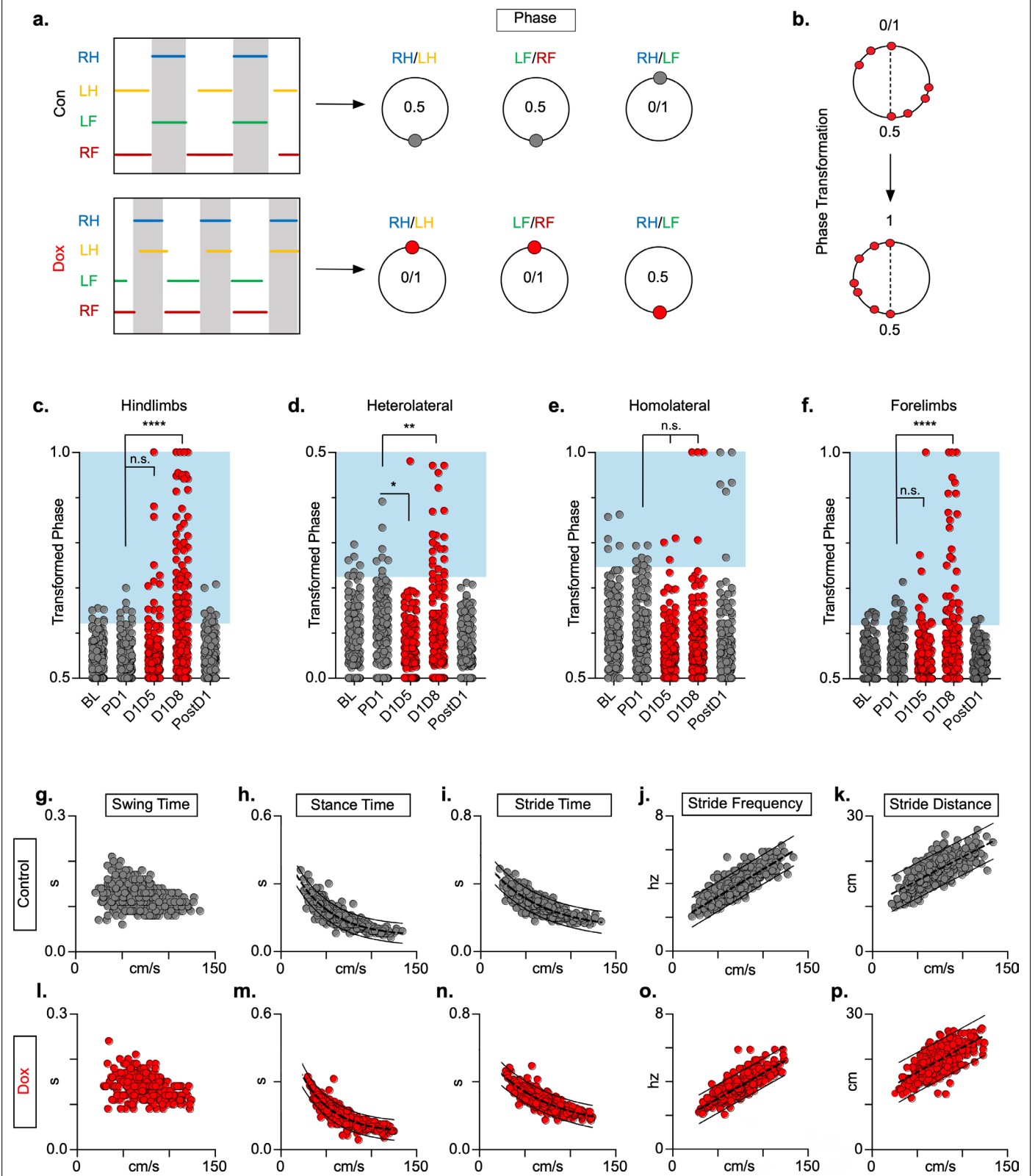

**Figure 2.** Silencing long ascending propriospinal neurons (LAPNs) disrupts interlimb coordination without affecting key features of locomotion. (**a**) Representative footfall graphs are shown with corresponding coordination phase values for both control and doxycycline (Dox) timepoints (RH = right hindlimb, LH = left hindlimb, LF = left forelimb, RF = right forelimb). Shaded regions indicate swing phase of LH and RF within the footfall cycle. Phase values generated from footfall graphs were converted to a scale of 0–1 to 0.5–1 for the hindlimb, forelimb, and homolateral hindlimb-forelimb pairs

*Figure 2 continued on next page*

*Figure 2 continued*

and 0–0.5 for the heterolateral hindlimb-forelimb pair so they could be further examined on a linear scale (**b**). The hindlimb and forelimb pairs were significantly altered during LAPN silencing, while the heterolateral hindlimb-forelimb pair was mildly affected and the homolateral hindlimb-forelimb pairs remained unaffected (**c–f**), # steps beyond control variability: PD1 hindlimbs n = 7/168 [4.17%] vs. D1D5 hindlimbs n = 14/166 [8.43%]; n.s., z = 1.62; PD1 hindlimbs n = 7/168 [4.17%] vs. D1D8 hindlimbs n = 72/161 [44.72%]; p < 0.001, z = 9.63; PD1 heterolateral n = 8/168 [4.76%] vs. D1D5 heterolateral n = 1/166 [0.60%]; p < 0.05, z = 2.38; PD1 heterolateral n = 8/168 [4.76%] vs. D1D8 heterolateral n = 21/161 [13.04%]; p < 0.01, z = 2.65; PD1 homolateral n = 7/168 [4.17%] vs. D1D5 homolateral n = 3/166 [1.80%]; n.s., z = 1.27; PD1 homolateral n = 7/168 [4.17%] vs. D1D8 homolateral n = 4/161 [2.48%]; n.s, z = 0.85; PD1 forelimbs n = 14/168 [8.33%] vs. D1D5 forelimbs n = 12/166 [7.23%]; n.s., z = 0.38; PD1 forelimbs n = 14/168 [8.33%] vs. D1D8 forelimbs n = 35/161 [21.74%]; p = 0.001, z = 3.45; binomial proportion test; circles = individual step cycles; shaded region = values beyond control variability. Spatiotemporal measures (swing time, stance time, stride time, stride distance) were plotted against speed for control (**g–k**) and Dox (**l-p**) timepoints. An exponential decay line of best fit is displayed for stance time and stride time graphs (stance time: control $R^2$ = 0.785 vs. Dox $R^2$ = 0.735; stride time: control $R^2$ = 0.708 vs. Dox $R^2$ = 0.667), while a linear line of best fit is displayed for stride distance (stride distance: control $R^2$ = 0.584 vs. Dox $R^2$ = 0.513; line of best fit indicated by dotted line); 95% prediction intervals are also shown as solid lines.

The online version of this article includes the following figure supplement(s) for figure 2:

**Source data 1.** Silencing long ascending propriospinal neurons (LAPNs) disrupts interlimb coordination without affecting key features of locomotion.

indicating alternation, while the heterolateral hindlimb-forelimb pair would concentrate around 0 or 1, indicating synchrony (***Figure 2a***). To eliminate any discrepancies between lead limb selection in the animals, phase values were converted from a circular scale (0–1) to a linear scale (0.5–1) (***Figure 2b***).

At control timepoints, hindlimb and forelimb pairs maintained a phase value around 0.5 (***Figure 2c and f***, gray; ***Figure 2—source data 1***). As expected, phase values for the heterolateral and homolateral hindlimb-forelimb pairs focused around synchrony and alternation, respectively, although with greater variability (***Figure 2d and e***, gray; ***Figure 2—source data 1***). We determined the mean phase value of the limb pairs and any value >2 SD from this mean was considered 'irregular', as indicated by the blue boxes (***Figure 2***, c-f). Silencing LAPNs disrupted left-right alternation of the hindlimb and forelimb pairs, with modest changes to the heterolateral hindlimb-forelimb pair (***Figure 2***, red; ***Figure 2—source data 1***). The homolateral limb pair remained unaffected (***Figure 2e***, red; ***Figure 2—source data 1***). The effects of silencing on limb pair relationships were reversed when Dox was removed from the drinking water (***Figure 2c, d and f***, 'PostD1'; ***Figure 2—source data 1***). However, silencing did not affect the speed-dependent gait indices of swing time (***Figure 2g and l***; ***Figure 2—source data 1***), stance time (***Figure 2h and m***; ***Figure 2—source data 1***), stride time (***Figure 2i and n***; ***Figure 2—source data 1***), stride frequency (***Figure 2j and o***; ***Figure 2—source data 1***), or stride distance (***Figure 2k and p***; ***Figure 2—source data 1***). The excursion of the proximal limb angle remained unaffected by silencing, whereas there were slight, but significant effects on the excursion of the distal limb angle (***Figure 1—figure supplement 1j***; ***Figure 1—figure supplement 1—source data 1***; ***Pocratsky et al., 2020***). These findings replicated our earlier observations and showed that LAPNs secure interlimb coordination with little to no change in intralimb coordination or other fundamental gait characteristics in uninjured rats.

## Spared LAPNs express synapse-silencing eTeNT.EGFP chronically

The majority of LAPN axons reside within the VLF (***English et al., 1985***; ***Molenaar and Kuypers, 1978***; ***Reed et al., 2006***). Importantly, we found the average spared white matter at the injury epicenter to be approximately 19% (***Figure 3a***; ***Figure 3—source data 1***) and that the outermost rim of the VLF was largely spared in this injury model (***Figure 3b–k***; ***Figure 3—source data 1***).

During Dox administration, eTeNT.EGFP is produced allowing for verification of LAPN projections in the cervical enlargement via immunohistochemical amplification of EGFP (***Figure 3***; ***Figure 3—source data 1***). Amplification revealed EGFP-positive puncta (putative axon terminals) surrounding neuronal processes/somata in the caudal cervical enlargement (***Figure 3l and m***; ***Figure 3—source data 1***). Similar to uninjured histology (***Pocratsky et al., 2020***), eTeNT.EGFP co-localized with synaptophysin (***Figure 3o and p*** ***Figure 3—source data 1***), vesicular GABA transporter (***Figure 3r***, VGAT; ***Figure 3—source data 1***), and vesicular glutamate transporter 2 (***Figure 3s***, VGlut2; ***Figure 3—source data 1***), markers of synapses and excitatory and inhibitory neurotransmitters, respectively. Isotype controls revealed minimal-to-no immunoreactivity (***Figure 3n and q***; ***Figure 3—source data 1***).

In rostral lumbar spinal cord segments, eTeNT.EGFP-positive LAPN cell bodies co-localized with fluorescent Nissl-stained (NeuroTrace) neurons in the intermediate gray matter (***Figure 3t–v***;

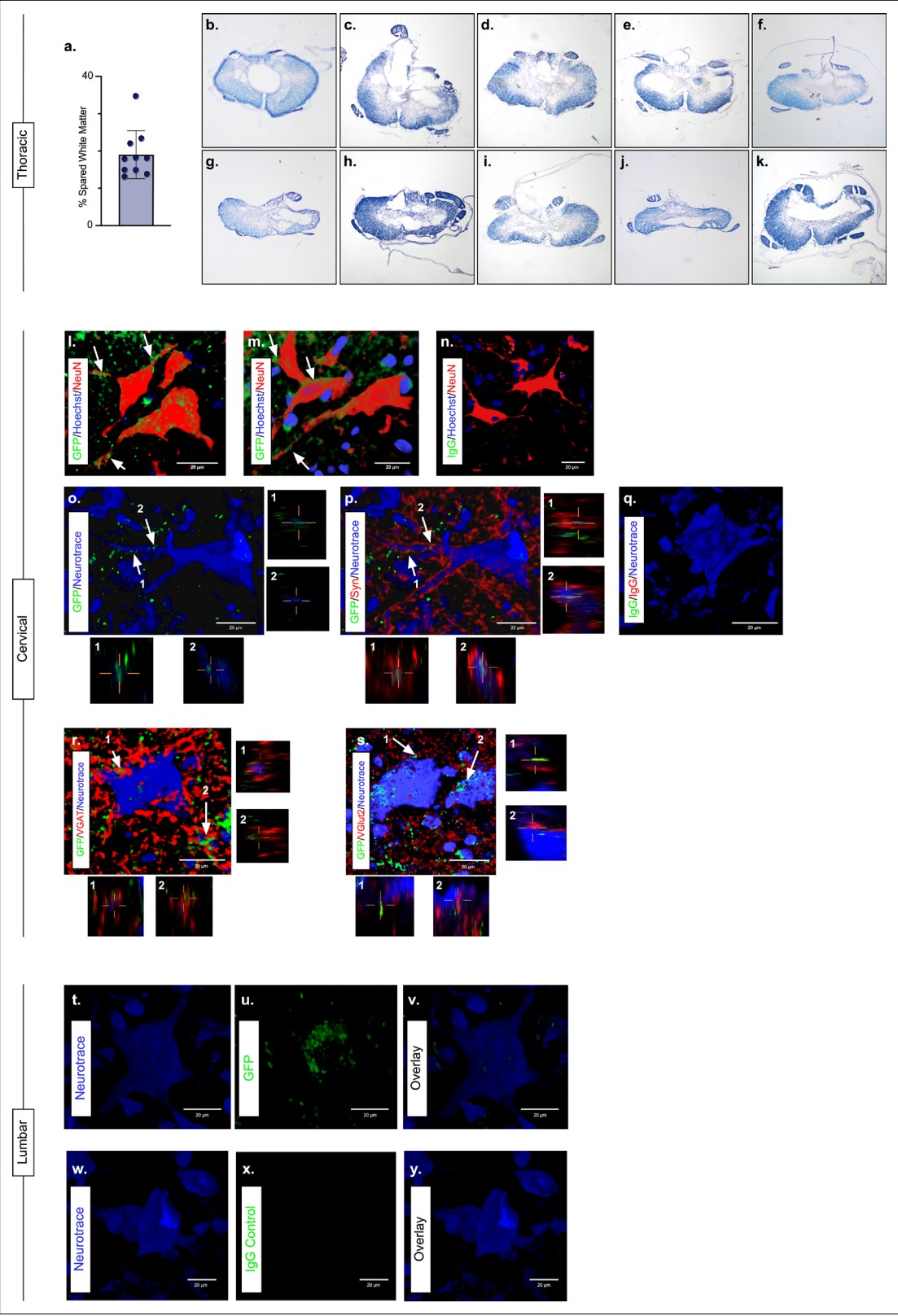

**Figure 3.** Presence of eTeNT.EGFP in putatively silenced long ascending propriospinal neurons (LAPNs) across the level of injury. In the thoracic spinal cord, the percentage of spared white matter at the epicenter ranges from 13.01% to 34.64% (**a**). White matter damage at the spinal cord injury epicenter as confirmed by histology (**b–k**). Individual images represent the injury epicenter of each animal used in the main data set (N = 10; average white matter percentage: 19.02%, standard deviations (SD) 6.44%). High magnification, volume-rendered images (**l-y**) involved full z-stacks (Amira) that ranged from

*Figure 3 continued on next page*

Figure 3 continued

55 to 64 slices at 0.45 μm optical steps (24.75–28.8 μm z-volume projection). Insets (**o–s**) show one 0.45 μm optical slice. These images demonstrate eTeNT.EGFP putatively positive fibers (green) surrounding NeuN-stained neurons (red) with Hoechst nuclear counterstain (blue) in cervical spinal cord segments of interest (**l,m**; C6-C7 spinal cord). White arrows indicate areas of co-localization. Isotype control reveals minimal immunoreactivity (**n**, IgG controls for eTeNT.EGFP shown). eTeNT.EGFP (green) signal co-localizes with neuronal processes (blue) and synaptophysin (red) (**o,p**). Isotype controls further show minimal reactivity (**q**, IgG controls of synaptophysin and eTeNT.EGFP shown). eTeNT.EGFP also co-localizes with the inhibitory neurotransmitter marker vesicular GABA transporter (**r**, VGAT, red) and the excitatory neurotransmitter vesicular glutamate transporter 2 (**s**, VGlut2, red). eTeNT.EGFP putative cell bodies (green) in the lumbar spinal cord co-localized with NeuroTrace fluorescent Nissl-stained neurons (blue) (**t–v**, L1-L2 spinal cord). Minimal presence of eTeNT.EGFP signal in isotype controls (**w–y**, L1-L2 spinal cord).

The online version of this article includes the following figure supplement(s) for figure 3:

**Source data 1.** Presence of eTeNT.EGFP in putatively silenced long ascending propriospinal neurons (LAPNs) across the level of injury.

*Figure 3—source data 1*; *Pocratsky et al., 2020*; *Reed et al., 2006*). Isotype controls showed no immunoreactivity (*Figure 3w–y*; *Figure 3—source data 1*). Taken together, these data suggest that some proportion of LAPNs were spared after moderate T10 contusion injury and that doubly infected LAPNs maintained both cell body and axon terminal-induced expression of eTeNT.EGFP following SCI. Thus, any behavioral changes seen during Dox administration would be concomitant with active eTeNT.EGFP expression in these spared LAPNs. Given the presence of eTeNT.EGFP in the tissue, previous work showing a similar dual-viral system infected >90% of LAPNs (*Brown et al., 2021*), and the high catalytic activity of eTeNT (*Montecucco and Schiavo, 1994*), it is likely that >90% of LAPNs localized to L2 and C6 were silenced here.

## Silencing LAPNs post-SCI improves locomotor function

We next explored the effects of LAPN silencing on locomotion after injury. We hypothesized that LAPNs spared after SCI contribute to spontaneous functional recovery, and that silencing them effectively makes the injury more severe and should reduce recovered locomotor function. We first evaluated the regularity index (RI), central pattern index (CPI), and plantar stepping index (PSI) as three related measures that rely on paw placement order to assess interlimb coordination. RI is simply the proportion of correctly patterned steps, excluding dorsal steps and irregularly patterned steps giving a sensitive measure of stepping quality after SCI (*Hamers et al., 2001*). In contrast, CPI evaluates steps in a rolling fashion allowing it to detect double steps, and it includes dorsal steps (*Patel et al., 2014*). Finally, PSI is the ratio of hindlimb steps over forelimb steps and is a direct assessment of forelimb and hindlimb coordination (see Materials and methods section for derivation of these measures) (*Kuerzi et al., 2010*; *Magnuson et al., 2009*). All three locomotor indices were modestly improved with LAPN silencing post-SCI (*Figure 4a–e*; *Figure 4—source data 1*), signifying a general improvement in paw placement order and timing (*Figure 4e*; *Figure 4—source data 1*).

In addition, gross locomotor function was assessed using the Basso, Beattie, Bresnahan locomotor rating scale (BBB) (*Basso et al., 1996*). Consistent with previous studies, BBB scores plateaued mid-scale, 11–13 out of 21 (*Basso et al., 2002*; *Smith et al., 2006*) and increased modestly during silencing (*Figure 4f*; *Figure 4—source data 1*), with a greater proportion of scores for each hindlimb exceeding 13 (range 13–18; *Figure 4g*; *Figure 4—source data 1*), suggesting that weight support by the hindlimbs, interlimb coordination, and toe clearance were improved during silencing.

## LAPN silencing leads to modest improvements in intralimb coordination after SCI

We then examined hindlimb kinematics during overground stepping post-SCI and following LAPN silencing. After SCI, compromised muscle activation leads to dorsal stepping where animals bear weight on the dorsal surface of their toes or hindpaws (*Figure 5a*). Proximal and distal limb angle excursions were determined for plantar and dorsal steps throughout each locomotor bout. Angle excursions were reduced post-SCI during most locomotor bouts and intralimb coordination was disrupted such that peaks (maximal extension) and troughs (maximal flexion) no longer occurred simultaneously (*Figure 5b, d and e*; *Video 1*; *Figure 5—source data 1*). Conditional silencing of spared LAPNs improved the coordination of the proximal and distal hindlimb angles, such that the cyclic properties of each angle were restored (*Figure 5c, g and h*; *Video 2*; *Figure 5—source data 1*). However, the excursions of both distal and proximal hindlimb joint angles were only slightly improved

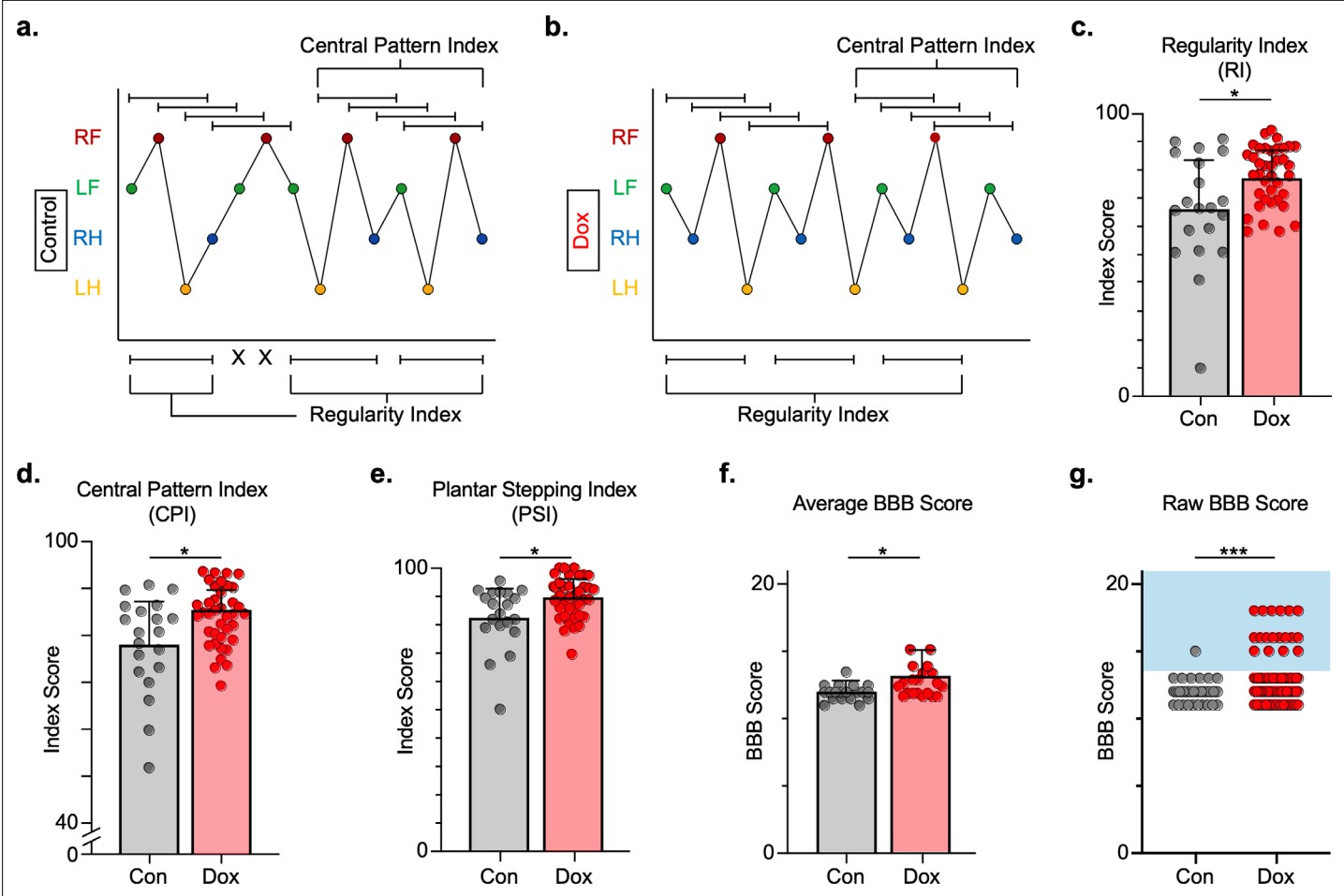

**Figure 4.** Silencing long ascending propriospinal neurons (LAPNs) post-spinal cord injury (SCI) restores coordination indices and improves gross locomotor outcomes. Footfall patterns are shown from example stepping passes to demonstrate how the gait indices regularity index (RI) and central pattern index (CPI) differed between post-SCI control (**a**) and post-SCI doxycycline (Dox) (**b**) stepping (RF = right forelimb, LF = left forelimb, RH = right hindlimb, LH = left hindlimb). 'X' indicates incorrect footfall order as defined by the typical foot patterns in RI. The group average RI scores (**c**, control RI: 65.74 ± 18.52 vs. Dox RI: 76.97 ± 7.70, t = 2.88, df = 9, p = 0.018; paired t-test), CPI scores (**d**, control CPI: 78.37 ± 8.39 vs. Dox CPI: 84.84 ± 3.39, t = 2.94, df = 9, p = 0.016; paired t-test), and PSI scores (**e**, control PSI: 82.72 ± 9.97 vs. Dox PSI: 89.07 ± 4.87, t = 2.76, df = 9, p = 0.022; paired t-test) are demonstrated. Averages are calculated from animal average indices with individual animal index scores shown from each control and Dox timepoint for transparency (gray circles and red circles for control and Dox, respectively). Average Basso, Beattie, Bresnahan locomotor rating scale (BBB) scores for control and Dox timepoints are shown (**f**, group average ± standard deviations [SD] [control to Dox]; p = 0.663, mixed model analysis of variance (ANOVA), Bonferroni post hoc. No significant difference was found between right and left BBB scores so they were combined for average and raw score ([left vs. right]; p = 0.001, mixed model ANOVA, Bonferroni post hoc; data not shown). To demonstrate BBB scores prior to averaging, right and left hindlimb raw BBB scores are shown in (**g**, control: n = 1/40 [0.025%] vs. Dox: n = 18/76 [23.68%]; p < 0.001, z = 4.23; binomial proportion test; circles = individual left or right BBB scores; blue shaded region = values beyond post-SCI control variability).

The online version of this article includes the following figure supplement(s) for figure 4:

**Source data 1.** Silencing long ascending propriospinal neurons (LAPNs) post-spinal cord injury (SCI) restores coordination induces and improves gross locomotor outcomes.

(*Figure 5f,i*; *Figure 5—source data 1*), suggesting that the primary effect of silencing may not be on intralimb coordination. These results suggest that changes seen during LAPN silencing after SCI involve modest improvements in the coordination and execution of the hindlimb joint movements.

## Hindlimb coupling and paw placement are improved during post-SCI silencing

We quantified dorsal steps in the ventral view based on the appearance of one or more toes, or the paw itself, being curled under to contact the walking surface (*Figure 6a*; *Keller et al., 2017*). We then

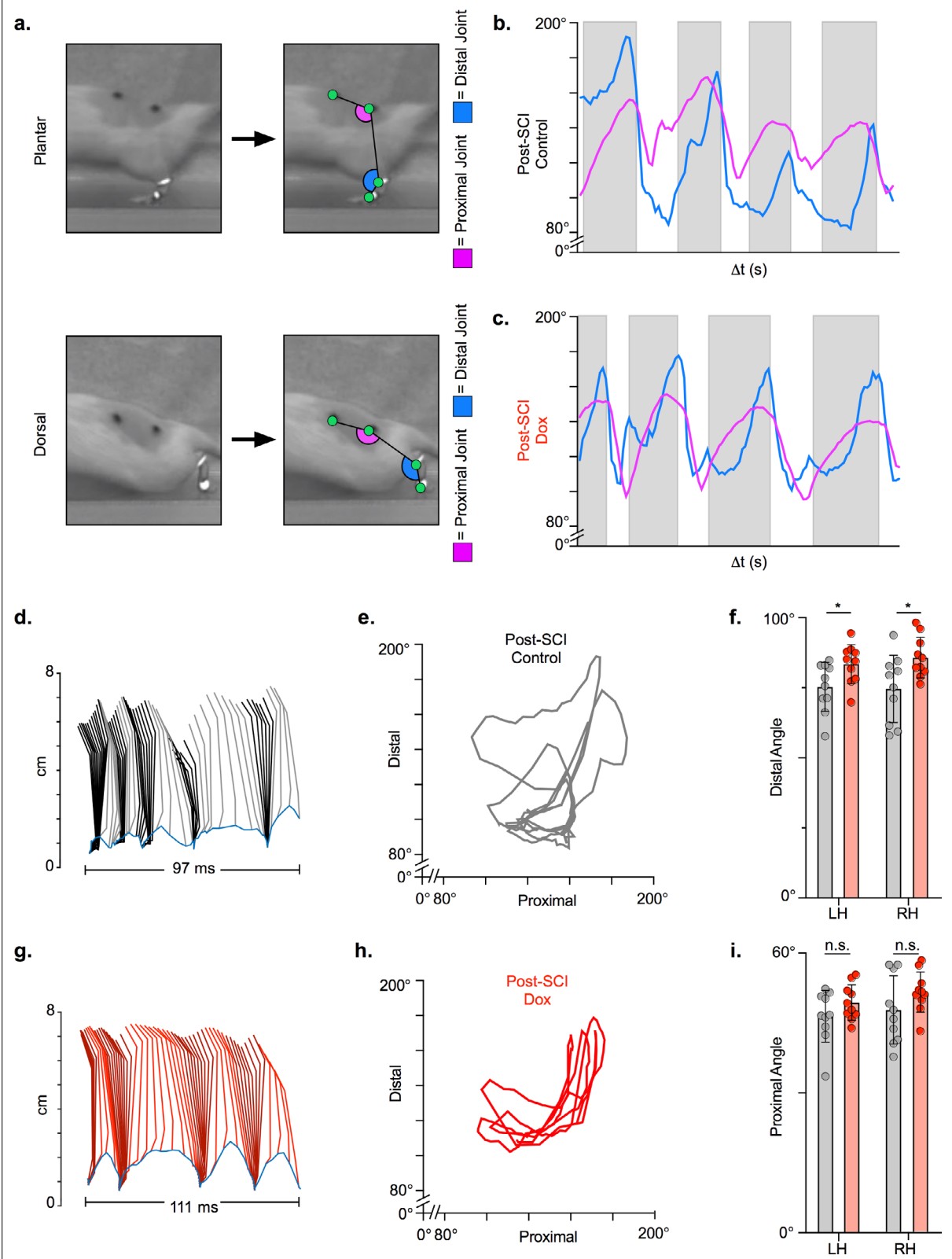

**Figure 5.** Long ascending propriospinal neuron (LAPN) silencing leads to modest improvements in intralimb coordination after spinal cord injury (SCI). A representative sagittal view of kinematic markers for plantar steps and dorsal steps is shown in (**a**). Excursion traces of the proximal (purple) and distal (blue) joint angles during post-SCI control (**b**) and doxycycline (Dox) (**c**) timepoints are shown. Two-dimensional stick figures of hindlimb stepping (**d,g**) and angle-angle plots (**e,h**) from the same passes shown in (**b,c**). Toe height throughout the pass is indicated as a blue trace on the bottom of the two-

*Figure 5 continued on next page*

*Figure 5 continued*

dimensional stick figures in (**d**) and (**g**), with the angles around 2 cm indicating the distal joint movements (ankle) and the angles around 6 cm indicating the proximal joint movements (hip). Gray boxes indicate periods in which the paw is in contact with the surface (stance phase). Angle-angle plots are shown to directly compare the distal joint angle to the proximal joint angle for the same recording frame. In (**f,i**), gray boxes indicate post-SCI control data while pink boxes indicate post-SCI Dox data. Minor improvements in distal joint angle excursion (**f**, control left: 74.62 ± 11.96 vs. Dox left: 85.74 ± 7.24, t = 3.35, df = 9, p = 0.009; control right: 75.29 ± 8.73 vs. Dox right: 83.40 ± 6.99, t = 2.70, df = 9, p = 0.024; paired t-tests; LH = left hindlimb, RH = right hindlimb) and proximal joint angle excursion (**i**, control left: 47.64 ± 7.33 vs. Dox left: 51.47 ± 4.33, t = 2.65, df = 9, p = 0.027; control right: 46.45 ± 5.58 vs. Dox right: 49.37 ± 3.80, t = 2.93, df = 9, p = 0.017; paired t-tests; bar indicates group average with individual animal averages shown as colored circles) are demonstrated as a result of silencing.

The online version of this article includes the following figure supplement(s) for figure 5:

**Source data 1.** Long ascending propriospinal neuron (LAPN) silencing leads to modest improvements in intralimb coordination after spinal cord injury (SCI).

compared the dorsal stepping index (DSI – the ratio of dorsal hindlimb steps to total hindlimbs steps) at post-SCI control and LAPN silenced timepoints. DSI was significantly reduced during silencing (11.21 ± 7.97) as compared to control timepoints (24.41 ± 16.1) (*Figure 6b*; *Figure 6—source data 1*). The sidedness of dorsal steps was unchanged between control and silencing, with ~60% of dorsal steps occurring in the right hindlimb and ~40% of dorsal steps occurring in the left hindlimb during both conditions (*Figure 6c*; *Figure 6—source data 1*). Thus, improvements seen during LAPN silencing were not biased to one side.

Using the left hindlimb as the reference limb, we examined the temporal coordination of the hindlimbs during overground stepping after SCI (*Figure 6d–g*; *Figure 6—source data 1*). Silencing improved left-right hindlimb coupling, as shown by an increase in the proportion of hindlimb steps that fell within the normal range of left-right phase values, concentrated around 0.5, which reversed when Dox was removed (*Figure 6e and f*; PD2, D2D5, D2D8; *Figure 6—source data 1*). The improved hindlimb coupling returned when Dox was administered a second time after injury (*Figure 6ef*; PD3, D3D8, D3D20; *Figure 6—source data 1*). To control for right-left bias due to behavioral asymmetry, we removed passes that included left hindlimb dorsal steps and re-examined interlimb coordination. Hindlimb coupling was still significantly improved when only plantar steps were considered, despite a reduction in the number of abnormally coordinated steps for both control and Dox timepoints (*Figure 6g*; *Figure 6—source data 1*).

Similar analysis of the heterolateral and homolateral hindlimb-forelimb pair revealed no significant improvement in coupling during silencing when dorsal steps were included (*Figure 6h and i*; red/pink/yellow; *Figure 6—source data 1*) or excluded (*Figure 6j*; *Figure 6—source data 1*). Finally, we assessed the right-left forelimb-forelimb coupling and found that it was unaltered throughout (*Figure 6k*; *Figure 6—source data 1*).

Three animals were removed from the data set prior to injury as they showed no perturbations of left-right alternation at any pre-injury Dox timepoint, presumably due to technical problems with the virus injections (*Figure 6—figure supplement 1a-i*; *Figure 6—figure supplement 1—source data 1*). Interestingly, these animals did not show silencing-induced improvements in

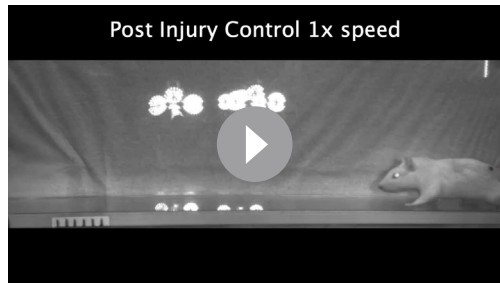

**Video 1.** Injured control example 'bad' pass from side camera view and ventral view (1× speed and 0.25× speed).
https://elifesciences.org/articles/70058/figures#video1

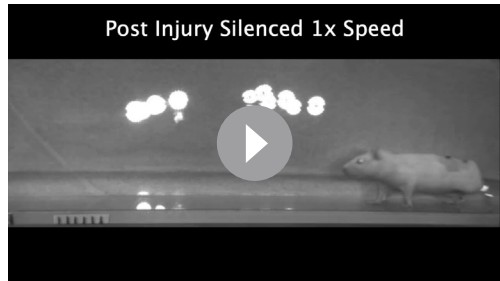

**Video 2.** Injured doxycycline (Dox) example 'good' pass from side camera view and ventral view (1× speed and 0.25× speed).
https://elifesciences.org/articles/70058/figures#video2

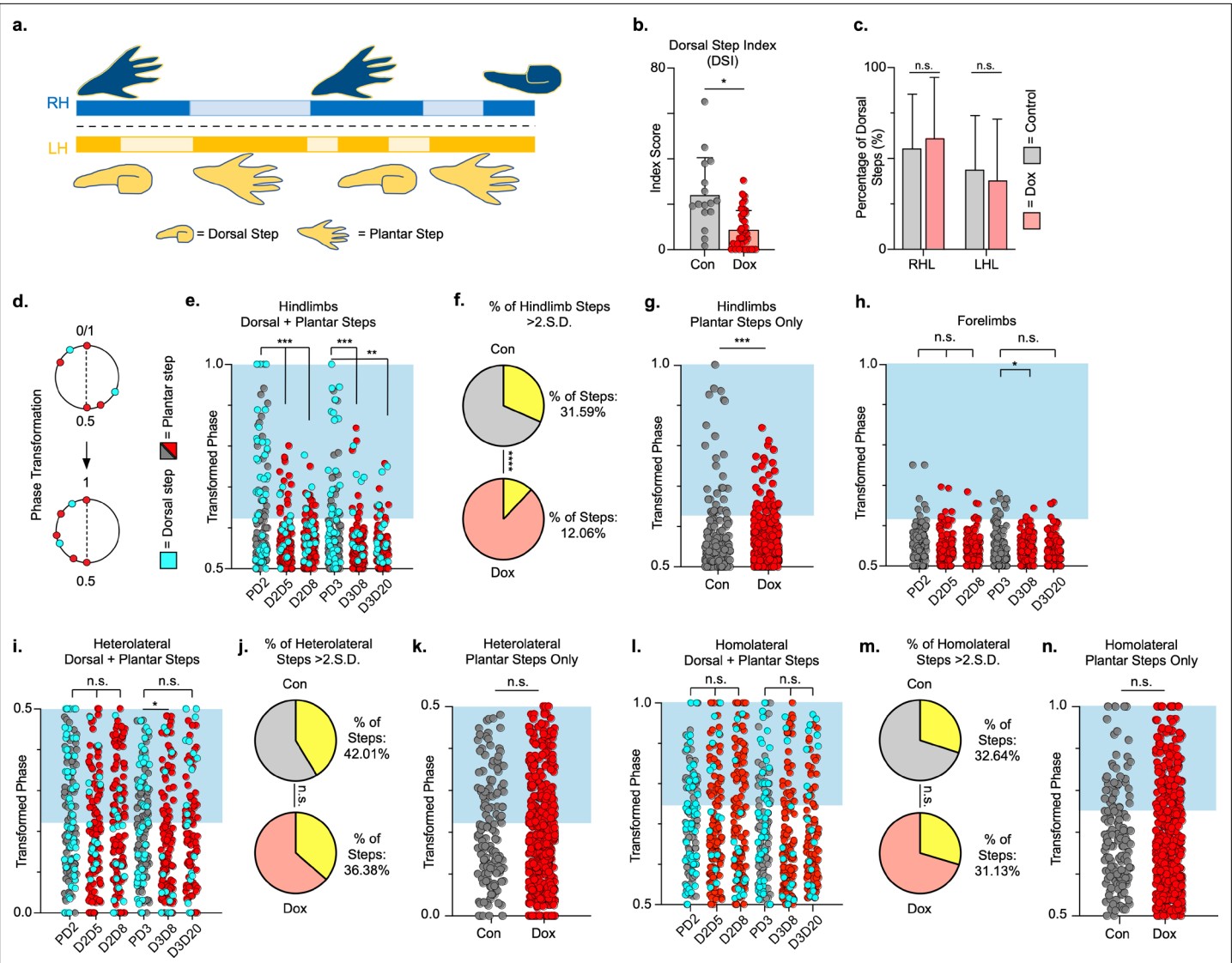

**Figure 6.** Hindlimb, but not hindlimb-forelimb, coupling relationships are restored during post-spinal cord injury (SCI) silencing. A representative pass with multiple dorsal and plantar steps shown in (**a**; LH = left hindlimb, light yellow boxes = LH swing phase, dark yellow boxes = LH stance phase; RH = right hindlimb, light blue boxes = RH swing phase, dark blue boxes = RH stance phase). Dorsal stepping index (DSI) is shown in (**b**, control DSI: 24.41 ± 14.69 vs. doxycycline (Dox) DSI: 11.75 ± 5.79; t = 3.383, df = 7, p = 0.012; individual circles represent individual animal scores at each post-SCI control and Dox timepoint). DSI accounts for total dorsal steps for both left and right hindlimbs. The total dorsal steps from (**b**) were separated based on sidedness: right hindlimb (RH) and left hindlimb (LH) (**c**, control right: 0.568 ± 0.247 vs. Dox right: 0.605 ± 0.238, t = .554, df = 7, p = 0.597; control left: 0.432 ± 0.247 vs. Dox left: 0.395 ± 0.238, t = .554, df = 7, p = 0.597; paired t-tests). The right hindlimb showed more dorsal steps overall; however, it maintained that percentage during Dox. No significant differences were seen in sidedness. As with pre-SCI data, circular phase was transformed from a scale of 0–1 to 0.5–1.0 to eliminate lead limb preferences (**d**). Transformed phase values plantar and dorsal steps for hindlimb pair (**e,f**, number of control hindlimb plantar and dorsal steps above 2 standard deviations [SD]: n = 91/288 [31.59%] vs. number of Dox hindlimb plantar and dorsal steps above 2 SD: n = 62/514 [12.06%]; p < 0.001, z = 7.05), heterolateral hindlimb-forelimb pair (**i,j**, number of control heterolateral limb plantar and dorsal steps above 2 SD: n = 121/288 [42.01%] vs. number of Dox heterolateral limb plantar and dorsal steps above 2 SD: n = 189/514 [36.77%]; n.s., z = 1.46), and homolateral hindlimb-forelimb pair (**l,m**, number of control homolateral limb plantar and dorsal steps above 2 SD: n = 94/288 [32.64%] vs. number of Dox homolateral limb plantar and dorsal steps above 2 SD: n = 160/514 [31.13%]; n.s., z = 0.44) are shown. See Supplemental data for additional binomial proportions tests. Plantar steps are indicated by gray circles (post-SCI control) and red circles (post-SCI Dox), while dorsal steps are indicated by teal circles for both post-SCI control and Dox data sets. The blue boxes indicate step phase values outside of normal variability (>2 SD) and are calculated from the specified uninjured limb pair control mean. The percentage of abnormal steps found above normal variability (circles within the blue boxes) was calculated for their respective limb pair (f,j,m, percentage of steps above 2 SD from e, i, and l, respectively, shown in yellow portion of pie graphs). Passes with any left hindlimb dorsal steps were removed and plotted for each of the aforementioned limb pairs (g,k,n: control hindlimbs plantar steps only: n = 33/157 [21.02%] vs. Dox hindlimbs without dorsal steps: n = 39/397 [9.82%]; p < 0.005, z = 3.13; control heterolateral limbs plantar steps

*Figure 6 continued on next page*

*Figure 6 continued*

only: n = 59/157 [37.58%] vs. Dox heterolateral limbs plantar steps only: n = 144/397 [36.27%]; n.s., z = 0.29; control homolateral limbs plantar steps only: n = 44/157 [28.02%] vs. Dox homolateral limbs plantar steps only: n = 121/397 [30.48%]; n.s., z = 0.58, B.P. tests). For the forelimb pair, no significant differences were seen between control and Dox phase during the first post-injury Dox administration (**h**; PD2 forelimbs: n = 7/143 [4.90%] vs. D2D5 forelimbs: n = 4/137 [2.92%]; n.s., z = 0.86; PD2 forelimbs: n = 7/143 [4.90%] vs. D2D8 forelimbs: n = 4/133 [3.00%]; n.s., z = 0.81, BP tests). Significance was detected between control and Dox at the D1D8 timepoint during the second Dox administration (PD3; D3D8), but no significance was found at the extended Dox timepoint (D3D20; PD3 forelimbs: n = 14/145 [9.66%] vs. D3D8 forelimbs: n = 2/135 [1.48%]; p < 0.05, z = 2.94; PD3 forelimbs: n = 14/145 [9.66%] vs. D3D20 forelimbs: n = 4/109 [3.67%]; n.s., z = 1.84, BP tests).

The online version of this article includes the following source data and figure supplement(s) for figure 6:

**Source data 1.** Hindlimb, but not hindlimb-forelimb, coupling relationships are restored during post-spinal cord injury (SCI) silencing.

**Figure supplement 1.** Animals excluded based on lack of behavioral outcomes pre-spinal cord injury (SCI) show no improvements in hindlimb coupling post-SCI.

**Figure supplement 1—source data 1.** Animals excluded based on lack of phenotype.

coordinated stepping post-SCI, suggesting that improvements in hindlimb stepping is an LAPN silencing-induced phenomenon (*Figure 6—figure supplement 1j-l*; *Figure 6—figure supplement 1—source data 1*).

## Key features of locomotion are improved during post-SCI LAPN silencing

To explore if silencing spared LAPNs post-SCI similarly improved speed-dependent spatiotemporal features of hindlimb stepping, we plotted swing time, stance time, and stride distance in relation to speed. These relationships were partially disrupted as a result of SCI, with both plantar and dorsal steps outside the normal range (*Figure 7a, b and d*; *Figure 7—source data 1*). Interestingly, measures that are typically associated with locomotor rhythm (stride time and stride frequency) were also disrupted (*Figure 7c and e*; *Figure 7—source data 1*), but these disruptions were typically associated with a dorsal step, which disrupts interlimb coordination. Silencing spared LAPNs improved these spatiotemporal relationships, reduced the number of abnormal steps, and reduced variability overall, regardless of dorsal or plantar stepping (*Figure 7f–j*; *Figure 7—source data 1*). Average swing (*Figure 7k*; *Figure 7—source data 1*) and stance times (*Figure 7l*; *Figure 7—source data 1*) improved, with greater improvement in swing time. Interestingly, the fundamental characteristic of duty cycle (the ratio of stance time to total stride time) was unchanged after silencing (*Figure 7m*; *Figure 7—source data 1*), suggesting that the improvements in locomotion were focused on interlimb coupling rather than the speed-dependent stance and swing times. Finally, the average stepping speed was unchanged during post-injury silencing (*Figure 7n*; *Figure 7—source data 1*), indicating that increased speed was not a primary factor in the improved spatiotemporal relationships. Together with the previous findings, these data suggest that silencing LAPNs after SCI influenced multiple aspects of locomotion driven by interlimb coordination, but also included step cycle spatiotemporal relationships ultimately leading to improved stepping overall.

## Hindlimb coordination during swimming post-SCI

Swimming is a bipedal activity where the hindlimbs provide propulsion while the forelimbs steer (*Gruner and Altman, 1980*). During swimming, the limbs are unloaded, and proprioceptive and cutaneous feedback are different than during stepping. In contrast to our overground findings, silencing LAPNs in the uninjured animal had no effect on left-right hindlimb alternation or any other salient feature of swimming (*Figure 8a and b*; *Figure 8—source data 1*), suggesting that the circuitry responsible for alternation during swimming is lumbar autonomous and does not rely on information carried by the LAPNs. Hindlimb coordination during swimming is strongly disrupted by SCI and surprisingly was modestly improved during post-SCI silencing (*Figure 8c*; *Figure 8—source data 1*). These results suggest that the role of LAPNs in swimming is altered post-SCI, and that the ability of LAPNs to appropriately integrate incoming sensory information from the hindlimbs may play a role in this changed outcome.

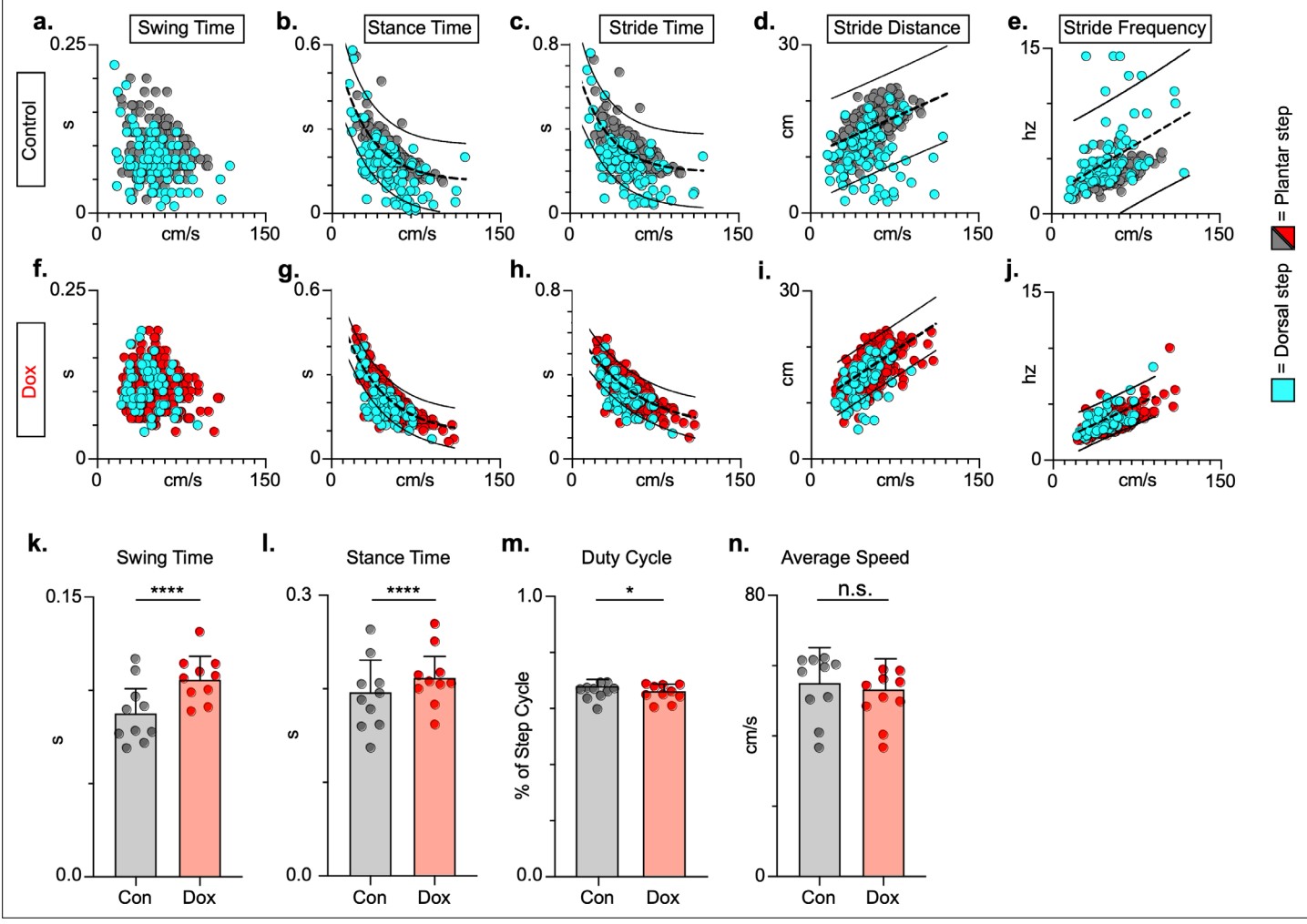

**Figure 7.** Key features of locomotion are improved following post-spinal cord injury (SCI) long ascending propriospinal neuron (LAPN) silencing. Relationships between swing time, stance time, stride time, and stride distance are plotted against speed for control (**a–e**) and doxycycline (Dox) (**f–j**) timepoints. Dorsal steps are indicated with teal circles for both control and Dox, while plantar steps are indicated with either gray or red circles for control and Dox, respectively. An exponential decay line of best fit is displayed for stance time and stride time graphs, while a linear line of best fit is displayed for stride distance (dotted line indicates line of best fit; stance time: control $R^2 = 0.433$ vs. Dox $R^2 = 0.656$; stride time: control $R^2 = 0.351$ vs. Dox $R^2 = 0.516$; stride distance: control $R^2 = 0.124$ vs. Dox $R^2 = 0.367$). 95% prediction intervals are shown for lines of best fit as solid lines. Average swing time (**k**, control swing time: $0.087 \pm 0.016$ vs. Dox swing time: $0.106 \pm 0.012$, t = 7.062, df = 9, p < 0.001, paired t-test) and average stance time (**l**, control stance time: $0.196 \pm .038$ vs. Dox stance time: $0.213 \pm 0.031$, t = 4.994, df = 9, p = .001, paired t-test) are indicated with circles representing individual animal averages. The average duty cycle (stance time/stride time) (**m**, control duty cycle: $0.679 \pm 0.028$ vs. Dox duty cycle: $0.663 \pm 0.030$, t = 2.678, df = 9, p = 0.025, paired t-test, significant due to tightness of data) and average speed (**n**, control speed: $55.17 \pm 9.20$ vs. Dox speed: $52.70 \pm 7.47$, t = 1.789, df = 9, p = 0.107, paired t-test) are plotted for control (gray) and Dox (red) timepoints with averages indicated by bars. One standard deviation (SD) is shown.

The online version of this article includes the following figure supplement(s) for figure 7:

**Source data 1.** Key features of locomotion are improved following post-spinal cord injury (SCI) silencing.

## Discussion

Consistent with our previous work, we found that silencing LAPNs in uninjured animals disrupted interlimb coordination of both the forelimbs and hindlimbs (*Pocratsky et al., 2020*). It is intuitive to hypothesize that any spared component of the inter-enlargement pathway represented by the LAPNs should participate in recovered locomotion following an SCI. Unexpectedly, silencing spared LAPNs post-SCI improved various aspects of locomotor function. These results demonstrate improvements in both intralimb and interlimb coordination when LAPNs are silenced, including improved

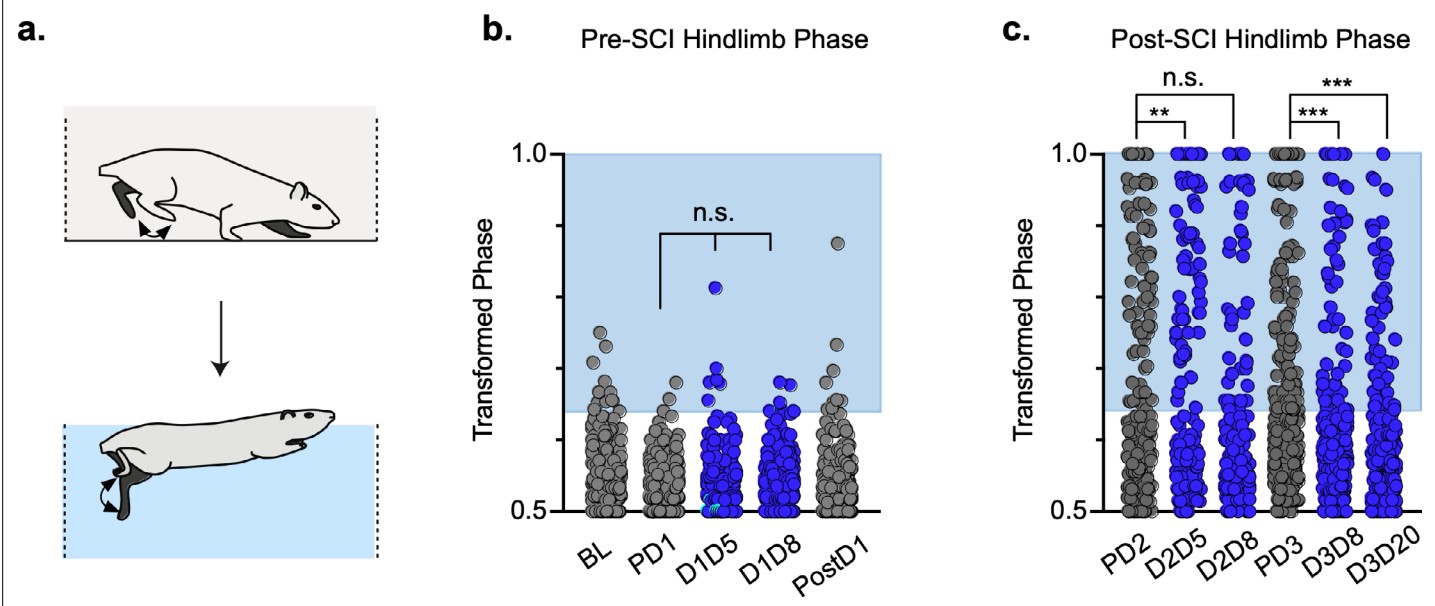

**Figure 8.** Interlimb coordination is improved during swimming. Similar to hindlimb coordination during overground locomotion, phase can be calculated between the hindlimbs during a swimming task (a). Hindlimb alternation was maintained during silencing of long ascending propriospinal neurons (LAPNs) in the pre-spinal cord injury (SCI) swimming task (b, PD1 hindlimbs n = 3/294 [1.02%] vs. D1D5 hindlimbs n = 6/302 [1.99%]; n.s., z = 0.97; PD1 hindlimbs n = 3/294 [1.02%] vs. D1D8 hindlimbs n = 7/278 [2.52%]; n.s., z = 1.37; gray circles indicate control swim kicks, blue circles indicate doxycycline [Dox] swim kicks). Post-SCI hindlimb coordination was disrupted as a result of injury (c, dark gray circles) and was modestly improved as a result of silencing (c, blue circles; PD2 hindlimbs n = 77/188 [40.95%] vs. D2D5 hindlimbs n = 43/166 [25.90%]; p = .003, z = 2.99; PD2 hindlimbs n = 77/188 [40.95%] vs. D2D8 hindlimbs n = 73/178 [41.01%]; n.s., z = 0.01; PD3 hindlimbs n = 113/281 [40.21%] vs. D3D8 hindlimbs n = 57/245 [23.27%]; p < 0.001, z = 4.15; PD3 hindlimbs n = 113/281 [40.21%] vs. D3D20 hindlimbs n = 51/233 [21.89%]; p < 0.001, z = 4.44; binomial proportions test). Blue boxes indicate values outside of normal variability for the pre-injury hindlimb pair mean > 2 standard deviations (SD).

The online version of this article includes the following figure supplement(s) for figure 8:

**Source data 1.** Interlimb coordination is improved during swimming.

paw placement order and timing and speed-dependent gait indices. Despite these improvements, hindlimb-forelimb coordination improved only slightly during post-SCI silencing.

These findings are counter-intuitive and suggest that the spared LAPNs interfere with hindlimb stepping following thoracic SCI. Given that few LAPNs have local axon collaterals (*Pocratsky et al., 2020*), these observations suggest that an ascending-descending, inter-enlargement loop involving LAPNs and the descending equivalents, LDPNs, may carry complementary temporal information between the two girdles. The thoracic contusion injury employed significantly reduces the cross-sectional area of white matter at the epicenter, while leaving the outermost rim of the lateral funiculus and VLF intact, precisely where the majority of long propriospinal axons reside. In addition to frank axon loss, demyelination occurs acutely, and some deficits in remyelination persist chronically, post-SCI (*Lasiene et al., 2008*; *Pukos et al., 2019*; *Totoiu and Keirstead, 2005*), which could lead to slowed and variable action potential conduction velocities. Thus, post-injury, the information carried by the partially spared LAPNs and LDPNs may no longer have the temporal precision necessary to reliably communicate the information needed for interlimb coordination. Rather than aiding in the recovery of stepping, it appears that spared LAPNs hinder the capability of the hindlimb locomotor circuitry below the lesion to function appropriately, contributing to diminished stepping capacity at chronic timepoints. We speculate that silencing LAPNs may remove excess 'noise' within the locomotor system, thereby increasing the ability of intrinsic lumbar circuitry to function independently. Alternatively, maladaptive plasticity may be occurring below the contusion site, leading to detrimental interactions between LAPN input and CPG circuitry.

Another interesting aspect to consider is the loss of influence of LAPNs on forelimb circuitry post-injury, suggesting that their role in providing temporal information helpful for forelimb coordination (alternation) is altered. In our recent study focused on LAPN silencing in uninjured animals, the

behavioral phenotype was strongly context-dependent, with disruptions to alternation occurring only during non-exploratory overground locomotion on a walking surface with good grip. Disruptions were insignificant during overground stepping on a slick surface, on a treadmill, or during exploratory-driven stepping (*Pocratsky et al., 2020*). These observations suggest a dynamic balance between spinal autonomy (where silencing disrupts alternation) and supraspinal oversight (where alternation is maintained). The parsimonious interpretation is that SCI results in increased supraspinal oversight of the cervical circuitry and forelimbs, preventing any disrupted flow of temporal information between the enlargements from disrupting forelimb function. In short, the influence of ascending circuitry post-SCI may be drastically reduced in the presence of increased supraspinal drive. Computational modeling could provide some insight into this conundrum as it would contribute to improved understanding of the drive between the hindlimbs and the forelimbs post-SCI.

Current literature has provided only a basic understanding of LAPN anatomy in the intact spinal cord (*Brockett et al., 2013*; *English et al., 1985*; *Reed et al., 2006*). Therefore, it is unclear whether anatomical and/or transmitter/receptor changes to LAPNs at either the level of the cell bodies in the lumbar cord or at the level of the axon terminals in the cervical cord are contributing to altered behavior after SCI. Further exploration of the anatomical profiles of this population of neurons will be essential to understand improvements seen during LAPN silencing. Another unresolved question is the potential for silencing-induced plasticity, as improvements in locomotor recovery during silencing were maintained for 20 days. Determining whether synaptic plasticity and/or anatomical changes in LAPNs play a role in improved stepping behaviors during or after silencing will further contribute to our understanding of these perplexing outcomes.

Alternatively, LAPNs may not undergo anatomical remodeling after SCI or silencing and any behavioral changes as a result of silencing could be due to maladaptive plasticity of sensory afferents. The context specificity of the LAPN silenced phenotype (*Pocratsky et al., 2020*) implies that LAPNs are integrators of hindlimb sensory information. As both proprioceptive (*Arber et al., 2019*; *Beauparlant et al., 2013*) and nociceptive (*Krenz and Weaver, 1998*) afferents caudal to the injury sprout post-SCI, newly formed circuits involving these afferents may include aberrant connections making it difficult for the LAPNs to properly integrate the information needed for interlimb coordination. Ex vivo studies suggest that the isolated lumbar and cervical enlargements are each capable of producing locomotor-like rhythms and, when in complete spinal preparations, that the lumbar circuitry has a greater influence on cervical circuitry than its reciprocal pathway (*Ballion et al., 2001*; *Juvin et al., 2012*; *Juvin et al., 2005*). If LAPNs are receiving aberrant sensory inputs from the hindlimb, removing them from the lumbar and lumbo-cervical circuitry could allow more autonomous correction of the intrinsic patterning within the lumbar cord.

A number of electrophysiologic studies have provided evidence that LAPNs are present in humans. For example, joint positions of the lower limbs can modulate reflexes in the upper limbs (*Baldissera et al., 1998*), stimulation of the peroneal nerve evokes short latency (<75 ms) responses in the upper limbs (*Zehr et al., 2001*) and stimulation of the lower limbs can trigger motor activity and reflexes in the upper limbs after SCI (*Butler et al., 2016*; *McNulty and Burke, 2013*). Clinically, epidural stimulation of the lumbosacral spinal cord improves both voluntary motor and locomotor performance in chronic SCI subjects (*Angeli et al., 2018*; *Angeli et al., 2014*; *Harkema et al., 2011*; *Minassian et al., 2013*). In large part, the mechanisms that underlie these outcomes are unknown, but clinically analogous stimulation in rodents was found to activate principally dorsal column and/or primary afferent sensory pathways (*Capogrosso et al., 2013*). It is unlikely that clinical epidural stimulation is directly activating lumbar locomotor circuitry or LAPNs, but likely influences lumbar motor circuitry via the more dorsally (posteriorly) located sensory circuitry. This leads us to speculate that the more successful stimulus parameters, chosen via painstaking epidural stimulus mapping (*Mesbah et al., 2017*), may be those that avoid activation of aberrant pathways, including any anomalous sensory input onto LAPNs or lumbar CPG circuitry. It will be critical to consider the presence of some spared pathways, such as this distinct population of LAPNs, as maladaptive to locomotor improvement in humans as it may be contributing to a ceiling effect for human locomotor recovery and to the beneficial effects of epidural and/or transcutaneous spinal cord stimulation.

Collectively, our results demonstrate that some spared axons may be detrimental to locomotor recovery after a thoracic contusion, contributing to a 'neuroanatomical-functional paradox' (*Fouad et al., 2021*), and raising the possibility that neuron- or axon-protective strategies leading to anatomical

sparing may not result in the expected benefits. Further studies utilizing reverse-engineering strategies to selectively silence/excite identified neuronal populations will be needed to identify which ascending and descending axons are contributory or inhibitory to recovery and the post-injury events leading to these drastic changes in functional role.

# Materials and methods

## Key resources table

| Reagent type (species) or resource | Designation | Source or reference | Identifiers | Additional information |
|---|---|---|---|---|
| Strain, strain background (*Rattus norvegicus*, females) | Sprague-Dawley rats | Envigo | #002 (RRID:MGI:5651135) | 200–220 g, approximately 10–12 weeks of age |
| Antibody | Anti-GFP (rabbit polyclonal) | abcam | ab290 (RRID:AB_303395) | IHC (1:5,000) |
| Antibody | Anti-VGAT (goat polyclonal) | Frontier Institute | VGAT-Go-Af620 (RRID:AB_2571623) | IHC (1:100) |
| Antibody | Anti-VGlut1 (guinea pig polyclonal) | Synaptic Systems | 135–304 (RRID:AB_887878) | IHC (1:500) |
| Antibody | Anti-VGlut2 (guinea pig polyclonal) | Synaptic Systems | 135–404 (RRID:AB_887884) | IHC (1:250) |
| Antibody | Anti-synaptophysin (guinea pig polyclonal) | Synaptic Systems | 101–004 (RRID:AB_1210382) | IHC (1:750) |
| Other | Neurotrace 640/660 | Fisher Scientific | 2047616 (RRID:AB_2572212) | IHC (1:50) Stain selective for Nissl substance. |
| Antibody | Anti-rabbit Alexa-Fluor Plus 488 (donkey polyclonal) | Invitrogen | A32790 (RRID:AB_2762833) | IHC (1:400) |
| Antibody | Anti-guinea pig Alexa-Fluor 594 (donkey polyclonal) | Jackson ImmunoResearch | #706-585-148 (RRID:AB_2340474) | IHC (1:200) |
| Antibody | Donkey anti-goat Alexa-Fluor 647 (donkey polyclonal) | Jackson ImmunoResearch | #705-605-147 (RRID:AB_2340437) | IHC (1:200) |
| Genetic reagent (*Rattus norvegicus*, females) | HiRet-TRE-EGFP.eTeNT | Generous gift from Tadashi Isa | n/a | Intraspinal injections of $1.6 \times 10^7$ vp/ml |
| Genetic reagent (*Rattus norvegicus*, females) | AAV2-CMV-rtTAV16 | Generous gift from Tadashi Isa | n/a | Intraspinal injections of $4.8 \times 10^{12}$ vp/ml |
| Genetic reagent (*Rattus norvegicus*, females) | HiRet-Cre | Generous gift from Zhigang He | n/a | Intraspinal injections of $1.6 \times 10^{12}$ vp/ml |
| Genetic reagent (*Rattus norvegicus*, females) | AAV2-CAG-FLEx-GFP | UNC Vector Core | Ed Boyden 100 µl – Control Vectors | Intraspinal injections of $3.5 \times 10^{12}$ vp/ml |
| Chemical compound, drug | Non-immune sera (rabbit IgG) | Jackson ImmunoResearch | #711-005-152 (RRID:AB_2340585) | IHC (1:5,000) |
| Chemical compound, drug | Sylgard | Dow Corning | Sylgard 184 Silicone Elastomer Kit | Applied to walkway chamber |
| Software, algorithm | SPSS | IBM | RRID:SCR_002865 | Version 22 |
| Software, algorithm | SigmaPlot | Systat | RRID:SCR_003210 | Version 11, used to generate polar plots |
| Software, algorithm | MaxTRAQ | Innovision Systems | n/a | Motion capture analysis |
| Software, algorithm | Matlab | Mathworks | | Custom-written code to analyze laminar distribution |
| Software, algorithm | Excel | Microsoft | | Custom-written code to analyze gait |

Experiments were performed in accordance with the Public Health Service Policy on Humane Care and Use of Laboratory Animals, and with the approval of the Institutional Animal Care and Use (IACUC) and the Institutional Biosafety (IBC) Committees at the University of Louisville.

Adult female Sprague-Dawley rats (215–230 g) were housed 2/cage under 12 hr light/dark cycle with ad libitum food and water. Each animal served as its own control. Power analysis for gait measures revealed that N = 6–10 could detect a true significant difference with power of 85–95%. A total of 16 animals entered the study to account for animal mortality following multiple surgical procedures. N = 4/16 animals were excluded for not meeting the a priori inclusion criteria based on uninjured behavioral phenotype during LAPN silencing. Of the remaining 12 animals, 2 died after surgery leaving N = 10 that were used for the main pre- and post-injury data set.

## Intraspinal injections of viral vectors to doubly infect LAPNs

Virus production, characterization, and injections were done as described previously (*Pocratsky et al., 2018*; *Pocratsky et al., 2017*; *Pocratsky et al., 2020*). Briefly, we utilized a dual viral vector technique to infect LAPNs via their terminals (C6) and cell bodies (L2, *Figure 1a and b*). HiRet-TRE-EGFP.eTeNT was injected at C6 and AAV2-CMV-rtTAV16 was injected at L2 as described previously (*Pocratsky et al., 2020*). In the presence of Dox, only doubly infected neurons that constitutively express the rtTAV16 Tet-On sequence induce expression of eTeNT, which is then transported anterogradely to the cell terminals. At the terminals, eTeNT prevents synaptic vesicle release, leading to 'silenced' neurotransmission that takes 3–5 days to develop once Dox is given. Removing Dox reverses the silencing and restores functional neurotransmission over a 3- to 5-day period both before and after SCI (*Figure 1e*; *Kinoshita et al., 2012*).

## Spinal cord injury

Two weeks after conclusion of uninjured Dox$^{ON}$ assessments, animals were re-anesthetized (ketamine:xylazine, 80 mg/kg:4 mg/kg; Henry Schein Animal Health; Akorn Animal Health) and the T9/T10 spine was immobilized using custom-built spine stabilizers (*Hill et al., 2009*; *Zhang et al., 2008*). SCI was performed as previously described (*Magnuson et al., 2009*). Post-injury care during recovery included antibiotics, analgesics, daily manual bladder expression, and supplemental fluids as previously described (*Magnuson et al., 2009*). Animals were allowed to recover for ~6 weeks before any additional pre-DOX assessments.

## Experimental timeline

Doxycycline hydrochloride (Dox, 20 mg/ml; Fisher Scientific BP2653-5, Pittsburgh, PA) was dissolved in 3% sucrose and provided ad libitum for 8 days pre-injury and for 8–21 days starting 6 weeks post-injury. Dox solution was made fresh, replenished daily, and monitored for consumption. All behavioral assessments were performed during the light cycle portion of the day and concluded several hours before the dark cycle began.

Prior to SCI, behavioral assessments were performed before viral injections (baseline, 'BL'), preceding Dox-induced LAPN silencing (pre-Dox, 'PD1'), during Dox (Dox1$^{ON}$D5-D8, 'D1D5', 'D1D8'), and 10 days post-Dox ('PostD1'). Animals that met the a priori inclusion criteria based on the behavioral phenotype subsequently received SCI. Following SCI, pre-Dox and Dox$^{ON}$ timepoint assessments were performed twice (Dox2 and Dox3) to assess the reproducibility of the silencing-induced behavioral changes. Data shown are from pre-injury control and Dox timepoints ('pre-SCI') and post-injury control and Dox timepoints ('post-SCI'). Individual and group comparisons were made for control vs. Dox uninjured and control vs. Dox injured at each timepoint. Behavioral analyses began on DoxD5 and were repeated on DoxD8. Terminal behavioral assessments occurred on DoxD8 and DoxD21.

## Hindlimb kinematics and intralimb coordination

Acquisition and analysis of hindlimb kinematics were performed as previously described (*Pocratsky et al., 2017*; *Pocratsky et al., 2020*). Briefly, we digitized the movements of the hindlimb by marking the skin overlying the iliac crest (anterior rim of the pelvis), the hip (head of the greater trochanter), ankle (lateral malleolus) and toe (fifth metatarsophalangeal joint) and recording from two sagittally oriented cameras, using the MaxTraq software package (Innovision Systems Inc; Columbiaville, MI) to calculate limb kinematics in 3D. The knee was not marked as excessive movement of the joint under the skin results in inaccurate localization (*Boakye et al., 2020*). However, this model easily detects changes in intralimb kinematics (*Barbeau and Rossignol, 1987*; *Boakye et al., 2020*). Data were exported to a custom Microsoft Excel workbook and hindlimb angles were calculated for each

digitized frame. The temporal relationship between the proximal and distal hindlimb angles was calculated for the left and right hindlimbs independently using the peak-to-peak duration of the lead angle during a single step cycle. Stick figures (2D) were generated as previously described (*Pocratsky et al., 2017*; *Pocratsky et al., 2020*).

## Overground gait analyses

Overground gait analyses were performed as previously described (*Pocratsky et al., 2020*). Dorsal steps, defined by the dorsum of the hindpaw contacting the surface during stance, was considered a step if it maintained contact with the surface during the swing phase of the step cycle. Hindlimb steps were analyzed using the left limb as the reference. As a result, dorsal steps were identified only if they involved the left limb.

Interlimb phase was calculated by dividing the initial contact time of one limb by the stride time (initial contact to initial contact) of the other limb. Phase was represented as circular polar plots to demonstrate interlimb coordination regardless of lead limb and was converted to a linear scale (0.5–1.0) to eliminate any lead limb preferences. When plotted linearly, blue shaded boxes on graphs represent a threshold of >2 standard deviations (SD) as calculated from uninjured control average and SD.

## Open field locomotor assessments

Hindlimb function during overground locomotion was assessed using the BBB open field locomotor scale, and assessments were performed by trained individuals blinded to experimental timepoints (*Basso et al., 2002*; *Caudle et al., 2015*). Prior to injury, BBB assessments occurred at baseline, pre-Dox, and Dox timepoints. After SCI, BBB scores were acquired weekly beginning at 7 days post-injury until scores plateaued (~6 weeks' post-SCI). BBB testing occurred prior to Dox administration (pre-Dox) and on any days of kinematic testing.

## Stepping coordination indices

RI, CPI, and PSI were used to evaluate gross motor coordination. RI = ([number of normal step sequence patterns × 4]/the number of paw placements). RI excludes any dorsal steps or irregularly patterned steps (step cycles in which paw placements are not sequential for all four paws, i.e. does not follow the cruciate, alternate, or rotary stepping patterns seen in normal rats) (*Cheng et al., 1997*; *Hamers et al., 2001*) and thus is a sensitive measure of stepping quality after SCI (*Hamers et al., 2001*). CPI = (number of correctly patterned step cycles/the total number of step cycles). Unlike RI, CPI includes both dorsal and plantar steps (*Patel et al., 2014*). PSI = (number of hindlimb plantar steps/number of forelimb plantar steps). In an uninjured animal, the ratio of hindlimb to forelimb steps is 1:1, or a PSI score of 1.0 (*Magnuson et al., 2009*). This provides a measure of hindlimb plantar stepping as compared to forelimb stepping after SCI.

## Spatiotemporal indices of stepping

For each individual plantar and dorsal step, temporal and spatial measures were plotted against their instantaneous speed (centimeters/second): swing time (the time the limb is in the air from lift off to initial contact), stance time (the time the limb is in contact with the ground from initial contact to lift off), stride time (stance time + swing time), all in seconds, stride/step frequency (1/stride time), and stride distance (distance traveled per step, centimeters). Average swing, stance and stride times, and stride distances were determined using the individual step values independent of speed. The average speed was calculated from the instantaneous speeds of each step analyzed. Averages were generated for each animal and were plotted with the group average.

## Hindlimb phase during swimming

Swimming assessments used a maximum of three passes with three to six complete stroke cycles per pass, per animal in each direction (to the left and to the right) (*Pocratsky et al., 2020*). The position of the toe at peak limb extension was digitized for both hindlimbs and the time of peak extension of the nearside hindlimb was divided into the length of time for one complete stroke cycle of the reference opposite hindlimb. Values were transformed as described above and the proportion of strokes with phases > 2 SD from control mean were compared across time.

## Histological analysis

Animals were euthanized at one of two timepoints: D3D8 (N = 2) or D3D20 (N = 11) to determine if viral expression and behavioral changes would persist beyond 1 week of Dox administration. Histological analysis revealed no differences between D8 and D20. Therefore, all images are shown from D20 animals.

Animals were overdosed with sodium pentobarbital, followed by pneumothorax and transcardial perfusion with 0.1 M phosphate-buffered saline (PBS) (pH 7.4) followed by 4% paraformaldehyde in PBS. Spinal cords were dissected, post-fixed for 1.5 hr, and transferred to 30% sucrose for >4 days at 4°C. Spinal segments C5-C8, T8-T12, and T13-L3/L4 were dissected, embedded in tissue freezing medium, and stored at –20°C until they were cryosectioned at 30 µm.

EGFP.eTeNT expression in the cervical spinal cord was confirmed immunohistochemically. Sections were incubated at 37°C for 30 min, then rehydrated in PBS for 10 min (pH 7.4, room temperature) followed by a 60 min incubation in a blocking solution made of nine parts milk solution (bovine serum albumin [BSA]), 0.75 g of powdered skim milk, and 14.25 ml 0.1% of PBS with Tween 20 (PBST) and one part 10% normal donkey serum (NDS). This was followed by another 10 min wash in PBS and overnight incubation at 4°C in milk solution containing primary antibodies. The primary antibody milk solution contained a combination of rabbit anti-GFP (to enhance eTeNT.EGFP) and either guinea pig anti-synaptophysin (presynaptic marker), guinea pig anti-vesicular glutamate transporter 1 (VGlut1, sensory afferents), guinea pig anti-vesicular glutamate transporter 2 (VGlut2, excitatory synapses), or goat anti-vesicular GABA transporter (VGAT, inhibitory synapses). For information on primary antibodies, see Key resources table.

On day 2, tissue sections were washed several times at room temperature, alternating between PBS and 0.1% PBST, followed by a 1 hr incubation in a dark room in milk solution containing the following secondary antibodies: donkey anti-rabbit Alexa-Fluor Plus 488 (1:500), donkey anti-guinea pig Alexa-Fluor 594 (1:200), and donkey anti-goat Alexa-Fluor 594 (1:200) (Key resources table). Tissue sections were washed with PBS for 10 min and then incubated at room temperature with fluorescent Nissl (NeuroTrace 640/660 Deep Red, ThermoFisher N21483, dilution of 1:100) in PBS for 1 hr to stain neuronal cell bodies. Tissue was then washed with PBS for 2 hr and coverslipped using Fluoromount (Southern Biotechnology Associates, Inc; Birmingham, AL). The above procedure was repeated on lumbar spinal cord sections. Isotype matched IgG with identical protein concentration was used as a negative control (donkey anti-rabbit IgG; Jackson ImmunoResearch #711-005-152).

Fluorescent images were captured on an Olympus Fluoview 1000 confocal microscope. Tissue sections were viewed with an oil immersion 100× objective using 488, 543, and 647 lasers (Olympus; Center Valley, PA). Z-stacks ranged from 55 to 64 slices at 0.45 µm optical steps. Neurons within the intermediate gray matter were imaged for both cervical and lumbar sections. Images were analyzed using Amira software as described (*Pocratsky et al., 2017*; *Pocratsky et al., 2020*).

Spared white matter at the injury epicenter was assessed as described previously using sections stained with iron eriochrome cyanine and alkali differentiation (EC) (*Magnuson et al., 2005*). Briefly, slides were allowed to warm at room temperature for 60 min before being placed in a hydration gradient consisting of xylenes, ethanol, and distilled water. EC stain was applied to the slides for 10 min, followed by two short applications of distilled water to remove excess stain (10–15 s). Slides were placed in differentiating solution for 30 s before air drying overnight. On day 2, slides were placed in xylenes solution for 10 min and were coverslipped with Permount. Sections were imaged using a Nikon Eclipse E400 light microscope at 10× magnification. Spot Software (v.5.1) was used to format images.

## Statistical analyses

Statistical analyses were performed using the SPSS v22 software package from IBM. Additional references for parametric and non-parametric testing were used (*Hays, 1981*; *Ott et al., 1977*; *Siegel and Castellan, 1988*). Differences between groups were deemed statistically significant at p ≤ 0.05. Two-tail p-values are reported.

The binomial proportion test was used to detect significant differences in the proportion of coordination values beyond control threshold (>2 SD) for the raw and transformed values of interlimb coordination of various limb pairs prior to and post-SCI. It was also used to determine statistical significance for interlimb phase, raw BBB score differences, intralimb phase, dorsal steps as a percentage

of total steps, and percentage of categorically organized steps (anti-phase, out of phase, in phase). Statistical outliers were excluded when appropriate.

Circular statistics were performed on the stepping interlimb coordination data sets (*Pocratsky et al., 2017*; *Zar, 1979*). We primarily used the non-parametric two-sample $U^2$ test based on a previously described rationale (*Pocratsky et al., 2017*; *Pocratsky et al., 2020*). The null hypothesis tested was whether two timepoints have the same expression of coupling pattern (right-left phase), that is, phase values are concentrated around 0.5 or are distributed throughout the possible range.

Regression analyses to compare the slopes for the lines of best fit were performed on the speed vs. spatiotemporal gait indices data sets, including speed vs. swing, stance and stride times, and stride distance for both hindlimb and forelimb pairs before and after SCI. For regression analyses post-SCI, plantar and dorsal steps were included in the analysis and dorsal steps are shown in teal; 95% prediction intervals are indicated on the graphs by solid lines. Dorsal steps were never seen for the forelimbs; thus all forelimb regression analyses were performed on plantar steps only.

Mixed model analysis of variance (ANOVA) followed by Bonferroni post hoc t-tests (where appropriate) were used to detect a significant difference in the BBB scores based on sidedness and condition (i.e. control vs. Dox; data not shown,). No significant differences were observed between sidedness and within condition. Repeated measures ANOVA were used to ensure that the average number of steps analyzed per animal during uninjured and injured timepoints. No differences were detected (for all comparisons, $p < 0.05$).

Paired t-tests were used to detect significant differences in proximal and distal angle excursion for intralimb coordination, average gross stepping measures including RI, CPI, PSI, and BBB scores, average intralimb phase, percentage of dorsal step sidedness, average swing time, average stance time, average duty cycle, and overall average speed at control and Dox pooled timepoints, respectively.

## Acknowledgements

The authors thank Dr Tadashi Isa and Dr Akiya Watakabe for generously providing the viral vector plasmids, Russell M Howard for assistance in viral vector production, Jason E Beare for assistance in confocal imaging and image processing, and Christine Yarberry, Kariena Andres, and Alice Shum Siu for their surgical support and assistance with animal care.

## Additional information

### Funding

| Funder | Grant reference number | Author |
| --- | --- | --- |
| National Institutes of Health | R01 NS089324 | David SK Magnuson Scott R Whittemore |
| National Institutes of Health | P30 GM103507 | Scott R Whittemore |
| Kentucky Spinal Cord and Head Injury Research Trust | 13-14 | David SK Magnuson |
| National Institutes of Health | R01 NS112304 | David SK Magnuson Scott R Whittemore |

The funders had no role in study design, data collection and interpretation, or the decision to submit the work for publication.

### Author contributions

Courtney T Shepard, Conceptualization, Formal analysis, Investigation, Project administration, Visualization, Writing – original draft; Amanda M Pocratsky, Brandon L Brown, Formal analysis, Methodology, Visualization, Writing – review and editing; Morgan A Van Rijswijck, Amberley S Riegler, Data curation, Formal analysis, Visualization; Rachel M Zalla, Formal analysis, Investigation, Visualization; Darlene A Burke, Data curation, Formal analysis, Methodology; Johnny R Morehouse, Formal analysis,

Methodology, Visualization; Scott R Whittemore, Conceptualization, Funding acquisition, Supervision, Writing – review and editing; David SK Magnuson, Conceptualization, Funding acquisition, Writing – review and editing

## Author ORCIDs
Courtney T Shepard http://orcid.org/0000-0002-1975-6188
Amanda M Pocratsky http://orcid.org/0000-0002-2834-8565
Brandon L Brown http://orcid.org/0000-0002-1557-5566
Scott R Whittemore http://orcid.org/0000-0001-6437-7200
David SK Magnuson http://orcid.org/0000-0003-3816-3676

## Ethics
This study was performed in strict accordance with the recommendations in the Guide for the Care and Use of Laboratory Animals of the National Institutes of Health. All of the animals were handled according to approved institutional animal care and use committee (IACUC) protocols (19644) of the University of Louisville. The protocol was approved by the Institutional Animal Care and Committee of the University of Louisville (OLAW/PHS Assurance No. A3586-01). All surgery was performed under approved modes of anesthesia, and every effort was made to minimize suffering.

## Decision letter and Author response
Decision letter https://doi.org/10.7554/eLife.70058.sa1
Author response https://doi.org/10.7554/eLife.70058.sa2

## Additional files

### Supplementary files
• Transparent reporting form

### Data availability
All data generated and analyzed during this study are included in the manuscript and supporting files. Source Data files have been provided for all figures that present data, rather than experimental design. Figure 1-figure supplement 1, Figures 2 through 8 and Figure 6-figure supplement 1.

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
