## [Editor Report]

The authors used viral transduction of tetanus toxin to silence synaptic transmission of long ascending propriospinal neurons (LAPN) to understand the role they play in function following contusion injury to the spinal cord. Kinematic analysis revealed that silencing LAPN improved interlimb and intralimb coordination rather than worsening it, as expected. This unexpected finding demonstrates that roles of LAPN and other interneurons following spinal cord contusion should not be assumed to mirror function of these neurons in uninjured spinal circuitry.

---

## [Decision Letter]

**Decision letter after peer review:**

Thank you for submitting your article "Silencing long ascending propriospinal neurons after spinal cord injury improves hindlimb stepping in the adult rat" for consideration by *eLife*. Your article has been reviewed by 3 peer reviewers, including Christopher Cardozo as the Reviewing Editor and Reviewer #1, and the evaluation has been overseen by Mone Zaidi as the Senior Editor.

Essential revisions:

The common thread to the three reviews that revisions were needed to improve clarity and readability of the manuscript. Specific comments that must be addressed include:

1. The manuscript relies heavily on gait analysis and standard parameters in the field. Adding a brief description of what these are and defining abbreviations where these are not already included in the text would help the reader who is not familiar with gait analysis.

2. The virally transducer model system used for modulating synaptic transmission must be introduced prior to presenting results.

3. Did you account for paw or limb rotation? Discuss how paw or limb rotation may have influenced results.

4. Effects of silencing on behavioral outcomes are modest and should be stated more conservatively.

5. A more complete description of methods should be included and should address the specific questions raised by reviewers 2 and 3.

6. Please carefully revise the manuscript to address the specific points raised by reviewer 3 regarding clarity.

7. Please address point 3 from reviewer 3 regarding homolateral coupling.

8. Please consider including discussion of the alternative interpretations of the data (Point 5, reviewer 3).

*Reviewer #1 (Recommendations for the authors):*

The manuscript would benefit from careful proofreading for tense and word usage. There are some cases in which results are referred to in present rather than past tense.

The manuscript relies heavily on gait analysis and standard parameters in the field. Adding a brief description of what these are and defining abbreviations where these are not already included in the text would help the reader who is not familiar with gait analysis. A bit more precision in descriptions of findings in the Results would help the reader who is not a specialist in gait understand what the changes in gait parameters mean. For example, interlimb coordination may refer to forelimb-hindlimb or left-right hindlimb. One can get through the results and figure out what they mean but it is sometimes unclear from the text.

Results:

The results would be easier to understand if the experimental model system summarized in Figure 1 was introduced before getting into the details of the results. As an example, the reader has not idea of the rationale was for testing for eTeNT.EGFP for EGFP-positive fibers. One has to dig through the methods to understand what these abbreviations mean and what is being measured. Explaining the experimental design (summarized in Figure 1) and approach (use of these various viral constructs to ablate synaptic transmission in a very specific way that is tunable with doxycycline) would greatly help the reader.

*Reviewer #2 (Recommendations for the authors):*

Abstract should be improved. Can you be more specific reporting results? Many sentences are too broad and "whishy-washy" regarding results and implications.

Introduction

It would be useful to include further information about LDPNs. What are the known interactions between the descending and ascending LPNs? Why is there a need for both?

Results

As alluded previously, no knee marker: if markers are on joint, can only determine hip or foot displacements no hip/knee and knee/ankle as described. Linking hip to ankle marker gives false reading?

How did you account for paw or limb rotations? Medial and lateral rotations at the hip also pronation or supinations at the ankle could provide very distorted 2D representations of joint angles.

Again, as alluded previously, no toe marker results in no information about toe drag.

How many LAPNs were spared? Is there sprouting of terminals? Why was there not quantification of anatomical data?

Co-localization difficult to see in Figure 2

Were angular displacements calculated for both plantar and dorsal steps? And averaged? Why?

Swimming test: Did the injury induce rats to use their forelimbs? How did that influence results?

Discussion

Behaviour improvements with silencing are modest, some of the conclusions should be toned down? For example, intralimb coordination and other kinematic parameters measurements are incomplete.

How is this effect happening? Authors speculated that LAPNs have maladaptations after SCI, but no evidence.

No forelimb effect could simply be due to redundancies in the system. Also, no evidence for their contention for supraspinal oversight of forelimbs.

Discussion in humans: what is the evidence this pathway and mechanism is similar in humans, as humans are bipedal animals. Maybe the effect would be similar to the results from the swimming tests?

Methods

Brief descriptions of the methods are requirement for a basic understanding. For example, there was no indication whether kinematics measurements were taken on a treadmill, open field, catwalk, etc.?

Pg 14. What is "irregularly patterned steps"?

*Reviewer #3 (Recommendations for the authors):*1. Prior findings were context dependent (i.e. on 'grippy' surfaces but not smooth/slick, non-exploratory locomotion). There is no mention of context here. Is there a context dependency to these effect post-SCI or were the same surfaces/contexts used?

2. Some of the benchmarks for comparison were not well described and some of indices used were not explained fully enough to distinguish between them or to see how they may have different implications for function.

a. For the RI, there are discrepancies in the Methods and the more detailed description in the paper referred to. This is likely due to the wording, rather than the actual measure. According to the paper referred to, the denominator is number of paw placements, not cycles (there are 4 paw placements per cycle). The dorsal or irregularly patterned are not excluded, except from the numerator, so that they are 'counted against' the index.

b. Based on the explanation provided, it is not clear how the CPI is different from the RI. The CPI does not appear to be included in the references referred to. What is "correctly patterned"? Is it order or phasing of one foot relative to another (i.e. alternation)?

c. Plantar step and dorsal step indices are presented as different measures but they appear to be the inverse of each other (based on the numbers). The plantar step index is described as plantar hindlimb steps relative to forelimb steps. Is a plantar step index of less than 100 due the presence of dorsal steps? Or is this a measure of forelimb-hindlimb coordination deficits?

d. In Figure 6, an indication of the "normal range" that is being referred to (in a-c) would be helpful, or alternatively, defining what is considered abnormal. The swing and stance times are said to be "improved". The basis for this judgment is not obvious.

3. Figure 5 does not include homolateral coupling. Although this was unaffected in intact, the LAPNs are both ipsilaterally and contralaterally projecting so this is of interest. Whether or not there are differences in homolateral coupling, this could be discussed in terms of anatomical work of the authors with regards to the populations of LAPNs being silenced and the more caudal LAPNs which are presumably (?) unaffected by the silencing.

4. Relatedly, information related to efficacy of the technique (i.e. approximate % of the LAPN population affected, distance of spread within lumbar segments when L2 is targeted) would enhance the interpretation as well.

5. There are several possible explanations for the results offered which are feasible based on the presented data. It seems that these may be possible to narrow down based on tissue from these experiments. Specifically, a quantification of LAPN terminals (or non-somatic fluorescence) in lumbar and cervical cord in these SCI rats compared to that in the uninjured may suggest, first, the degree to which terminals are lost after SCI (or an approximate proportion that are spared) and, second, if there is significant sprouting at either level. Since lumbar terminals were sparse in the intact, are there terminals seen in lumbar cord after SCI? Presence or absence may lend support to either the long loop effects or local gain of aberrant connections.

---

## [Author Response]

Essential revisions:The common thread to the three reviews that revisions were needed to improve clarity and readability of the manuscript. Specific comments that must be addressed include:1. The manuscript relies heavily on gait analysis and standard parameters in the field. Adding a brief description of what these are and defining abbreviations where these are not already included in the text would help the reader who is not familiar with gait analysis.

Brief descriptions of the various gait analysis/parameters have been added to the “Silencing LAPNs post-SCI improves locomotor function” (lines 148-156) section of the results. Additionally, the derivations for each of these indices are now clearly given methods (lines 422-432).

2. The virally transducer model system used for modulating synaptic transmission must be introduced prior to presenting results.

An introduction to the dual viral system has been added to the Results section “Silencing alters interlimb coordination while other key features of locomotion are maintained” (lines 78-85). Additionally, the new Figure 1 outlines the dual viral silencing system.

3. Did you account for paw or limb rotation? Discuss how paw or limb rotation may have influenced results.

Paw rotation is a common characteristic of stepping post-SCI and was at least partly accounted for by our kinematic assessment which used 2 cameras placed at angles allowing the limb movements on one side of the animal to be quantified in 3 dimensions, with some limitations compared to multi-camera set-ups. The use of 2 cameras for 3D estimates is described in the section “Hindlimb kinematics and intralimb coordination”. We consider the accuracy sufficient to support the primary conclusions which rely heavily on measures and indices that would not be influenced by paw or limb rotation. Figure 5 and Figure 2 supplement 1 both show 2D stick figures derived from the 3D coordinates (from MaxTraq 3D) where the ankle-toe distance would decrease during paw rotation, however the calculated angle would not be affected because it is derived from the 3D coordinates.

4. Effects of silencing on behavioral outcomes are modest and should be stated more conservatively.

The language used throughout the Results and Discussion has been modified to state the outcomes more conservatively. However, as the correct statistical analyses were performed, we believe it is still appropriate to refer the results as “significant” if/when the statistical outcomes have a p-value <.05.

5. A more complete description of methods should be included and should address the specific questions raised by reviewers 2 and 3.

Methodological details have been added to the Results and Methods sections specifically addressing the viral-based silencing and kinematic assessments. In addition, Figure 1 has been modified to clarify these issues by illustrating the silencing system (Figure 1a), the injection strategy (Figure 1b), the kinematic and gait analysis (Figure 1c,d) and the experimental timeline (Figure 1e).

6. Please carefully revise the manuscript to address the specific points raised by reviewer 3 regarding clarity.

Comment 2 from Reviewer 3 has multiple parts and is addressed below following each of reviewer 3’s comment(s).

7. Please address point 3 from reviewer 3 regarding homolateral coupling.

Figure 6 (previously figure 5) now includes data for homolateral coupling.

8. Please consider including discussion of the alternative interpretations of the data (Point 5, reviewer 3).

This is addressed below following point 5 from Reviewer 3.

Reviewer #1 (Recommendations for the authors):The manuscript would benefit from careful proofreading for tense and word usage. There are some cases in which results are referred to in present rather than past tense.

We have adjusted the tenses used and further proofread the manuscript.

The manuscript relies heavily on gait analysis and standard parameters in the field. Adding a brief description of what these are and defining abbreviations where these are not already included in the text would help the reader who is not familiar with gait analysis. A bit more precision in descriptions of findings in the Results would help the reader who is not a specialist in gait understand what the changes in gait parameters mean. For example, intralimb coordination may refer to forelimb-hindlimb or left-right hindlimb. One can get through the results and figure out what they mean but it is sometimes unclear from the text.

To improve readability and provide context to the findings we have added brief descriptions of the various gait analysis/parameters used to the “Silencing LAPNs post-SCI improves locomotor function” section of the results (lines 148-158) and we added a new Figure 1 to illustrate our kinematic/gait assessment more clearly. Additionally, the derivations for each of the gait indices are now clearly given in the methods (lines 422–442).

Results:The results would be easier to understand if the experimental model system summarized in Figure 1 was introduced before getting into the details of the results. As an example, the reader has not idea of the rationale was for testing for eTeNT.EGFP for EGFP-positive fibers. One has to dig through the methods to understand what these abbreviations mean and what is being measured. Explaining the experimental design (summarized in Figure 1) and approach (use of these various viral constructs to ablate synaptic transmission in a very specific way that is tunable with doxycycline) would greatly help the reader.

We agree with the Reviewer and have added details to the “Silencing alters interlimb coordination while other key features of locomotion are maintained” section of the results (lines 78-85). Additionally, a new Figure 1 has been amended to provide further detail on the experimental setup and design.

Reviewer #2 (Recommendations for the authors):Abstract should be improved. Can you be more specific reporting results? Many sentences are too broad and "whishy-washy" regarding results and implications.

The abstract has been significantly rewritten to provide specifics about the behavioral (locomotor) outcomes and is now more direct about possible implications.

IntroductionIt would be useful to include further information about LDPNs. What are the known interactions between the descending and ascending LPNs? Why is there a need for both?

Direct interactions between the LAPNs and LDPNs are not known, but some additional information about these two populations has been added to the introduction (lines 41-44) and discussion (lines 250-262)

ResultsAs alluded previously, no knee marker: if markers are on joint, can only determine hip or foot displacements no hip/knee and knee/ankle as described. Linking hip to ankle marker gives false reading?

The Reviewer is correct, we do/did not mark the knee for intralimb kinematics as there is extensive movement of the knee joint under the skin (PMIDs:19270266, 14673005, 32605423). To accurately identify and track the knee the lengths of the tibia and femur are needed, and software is needed to triangulate knee position. Thus, our approach is to mark the Iliac crest, the head of the greater trochanter, the lateral malleolus of the ankle, and the most lateral metatarsophalangeal joint to create a three-segment, two-angle model of the limb that is both accurate and sensitive for detecting intralimb kinematics (PMIDs: 32605423, 29213073, 32902379). This has now been clearly stated in the methods section “Hindlimb kinematics and intralimb coordination” (lines 387-400)

How did you account for paw or limb rotations? Medial and lateral rotations at the hip also pronation or supinations at the ankle could provide very distorted 2D representations of joint angles.

Repeat from above: Paw rotation is a common characteristic of stepping post-SCI and was at least partly accounted for by our kinematic assessment which used 2 cameras placed at angles allowing the limb movements on one side of the animal to be quantified in 3 dimensions, with some limitations compared to multi-camera set-ups. The use of 2 cameras for 3D estimates is described in the section “Hindlimb kinematics and intralimb coordination”. We consider the accuracy sufficient to support the primary conclusions which rely heavily on measures and indices that would not be influenced by paw rotation. Figure 5 and Figure 2 supplement 1 both show 2D stick figures derived from the 3D coordinates (from MaxTraq 3D) where the ankle-toe distance would decrease during paw rotation, however the calculated angle would not be affected because it is derived from the 3D coordinates. We hope this clarifies our experimental approach and analysis.

Again, as alluded previously, no toe marker results in no information about toe drag.

The toe is marked (as the most lateral metatarsophalangeal joint), and we have verified that this is clearly stated in the methods (lines 388-395). This information is now illustrated clearly in the new Figure 1.

How many LAPNs were spared? Is there sprouting of terminals? Why was there not quantification of anatomical data?

These are excellent questions/points raised by the Reviewer but are beyond the scope of the current study. To correctly address these questions would require additional cohorts of animals and use of a dual-viral system specifically designed for anatomic quantification. The axons of long propriospinal neurons (spanning > 6 spinal levels) are positioned superficially in the lateral and ventrolateral white matter which increases the likelihood that substantial numbers of these axons remain anatomically intact post-SCI (PMID: 15514981).

Co-localization difficult to see in Figure 2

Figure 2 has been slightly adjusted for visibility.

Were angular displacements calculated for both plantar and dorsal steps? And averaged? Why?

Yes, we calculated intralimb coordination and kinematics for both plantar and dorsal steps and chose not to focus on just plantar steps because dorsal steps are one of the consequences of the injury delivered that does mildly influence limb kinematics. During silencing after injury the number of dorsal steps was significantly reduced and we believe this was reflected in the intralimb kinematic outcomes (like angular displacement). Thus, it made sense to us to include both dorsal and plantar steps. Figure 5 illustrates how dorsal stepping might influence angular displacement.

Swimming test: Did the injury induce rats to use their forelimbs? How did that influence results?

The injuries were mild enough for the animals to have hindlimb-only swimming when a functional plateau was achieved (albeit at a slower speed than normal). We choose swimming passes for analysis that are uninterrupted and where the animals are swimming down the middle of the lane, since both uninjured and injured animals (at their functional plateau, silenced or not) will use their forelimbs for steering and/or to push off the sides of the tank.

DiscussionBehaviour improvements with silencing are modest, some of the conclusions should be toned down? For example, intralimb coordination and other kinematic parameters measurements are incomplete.

The conclusions have been toned down where appropriate. Given that the metatarsophalangeal joint is marked, and that we have more fully described our kinematic and gait assessments that utilize a 3D estimation of the limb using three segments and two angles (please see response to Reviewer 2) that is both accurate and sensitive to changes in intralimb kinematics (PMID: 32605423), and that we utilized appropriate statistical analyses, we content that it is still appropriate to refer to the results as significant if p<.05. Furthermore, taken together, the improvements are not just statistically significant, but are functionally significant as can be seen in the included videos.

How is this effect happening? Authors speculated that LAPNs have maladaptations after SCI, but no evidence.

The Reviewer is correct. We do not have any evidence to help explain how/why these behavioral effects occur and but have presented several potential mechanisms in the discussion. Our laboratories are currently developing and optimizing the methods to narrow down these possibilities. As these methods must be empirically optimized for each application (PMID: 33828464), definitive answers about the mechanism(s) responsible for the behavioral outcomes are outside the scope of the current study.

No forelimb effect could simply be due to redundancies in the system. Also, no evidence for their contention for supraspinal oversight of forelimbs.

We have added to this section of the discussion in the hopes of supporting this speculation. In our previous paper (PMID: 32902379), we showed that LAPN silencing disrupted both forelimb and hindlimb alternation but in a context specific manner. We speculate that the context specificity is due to greater supraspinal oversight in some contexts as compared to others. Silencing the LAPNS after SCI did not disrupt forelimb alternation, suggesting that the context of SCI may involve greater supraspinal oversight over forelimb function. We agree that this could also simply be due to redundancies in the system, but what redundancies would be present in the SCI state, but not in the uninjured state?

Discussion in humans: what is the evidence this pathway and mechanism is similar in humans, as humans are bipedal animals. Maybe the effect would be similar to the results from the swimming tests?

Previously, we did not include human literature/references providing evidence for LAPNs in humans. This information has been added to the discussion (lines 310-329)

MethodsBrief descriptions of the methods are requirement for a basic understanding. For example, there was no indication whether kinematics measurements were taken on a treadmill, open field, catwalk, etc.?

We have clarified the context in which kinematics were done at the end of the introduction (lines 69-71), beginning of the results (lines 86-94) and in the methods (lines 388-400). Additional details about the viral system (lines 78-85) and locomotor indices have also been added (lines 148-156) to improve readability and comprehension. Finally, we have added a new Figure 1 to more clearly illustrate how we assess kinematics/gait.

Pg 14. What is "irregularly patterned steps"?

A clear definition of “irregularly patterned steps” has been added to the “Stepping coordination indices” in the methods (lines 424-427).

Reviewer #3 (Recommendations for the authors):1. Prior findings were context dependent (i.e. on 'grippy' surfaces but not smooth/slick, non-exploratory locomotion). There is no mention of context here. Is there a context dependency to these effect post-SCI or were the same surfaces/contexts used?

Thank you for bringing this to our attention. The context specificity of the previous findings and rationale for the current study design have been added to the end of the introduction (lines 69-71).

2. Some of the benchmarks for comparison were not well described and some of indices used were not explained fully enough to distinguish between them or to see how they may have different implications for function.

Brief descriptions of the various gait analysis/parameters used have been added to the “Silencing LAPNs post-SCI improves locomotor function” section of the results (lines 148-156). Additionally, the derivations for each of these indices are now clearly given methods (lines 422-432)

a. For the RI, there are discrepancies in the Methods and the more detailed description in the paper referred to. This is likely due to the wording, rather than the actual measure. According to the paper referred to, the denominator is number of paw placements, not cycles (there are 4 paw placements per cycle). The dorsal or irregularly patterned are not excluded, except from the numerator, so that they are 'counted against' the index.

A brief description for the interpretation of RI has been added to “Silencing LAPNS post-SCI improves locomotor function” section of the results (lines 148-156), and the equation for RI has been corrected in the methods section (lines 423-428).

b. Based on the explanation provided, it is not clear how the CPI is different from the RI. The CPI does not appear to be included in the references referred to. What is "correctly patterned"? Is it order or phasing of one foot relative to another (i.e. alternation)?

The difference between CPI and RI is now clearly stated in the Results section (148-156) and further details for both measures have been added to the methods (lines 422-432).

c. Plantar step and dorsal step indices are presented as different measures but they appear to be the inverse of each other (based on the numbers). The plantar step index is described as plantar hindlimb steps relative to forelimb steps. Is a plantar step index of less than 100 due the presence of dorsal steps? Or is this a measure of forelimb-hindlimb coordination deficits?

While the nomenclature is similar for these locomotor indices, they provide differing information related to locomotion and are not the inverse of one another. The Reviewer is correct, that a PSI of <1.0 (or <100%) would be due to the presence of dorsal hindlimb steps and based on its derivation is also a measure of forelimb-hindlimb coordination. Details about the interpretation and derivation of DSI and PSI have been added to the results (lines 182-183) and methods sections (lines 422-432) and are also provided below.

◾Dorsal stepping index (DSI) is defined as “the ratio of dorsal steps to total hindlimb steps” and assesses precision of stepping or paw placement of the hindlimbs.

– DSI = (# hindlimb dorsal steps / # total hindlimb steps)

◾Plantar stepping index (PSI) is defined as “the ratio of hindlimb plantar steps to number of forelimb steps” and assesses coordination of the forelimbs and hindlimbs.

– PSI = (# hindlimb plantar steps / # forelimb plantar steps)

d. In Figure 6, an indication of the "normal range" that is being referred to (in a-c) would be helpful, or alternatively, defining what is considered abnormal. The swing and stance times are said to be "improved". The basis for this judgment is not obvious.

The normal trend is indicated in Figure 2g-k, as this represents the expected spatiotemporal relationship in the same animals prior to SCI and without silencing.

3. Figure 5 does not include homolateral coupling. Although this was unaffected in intact, the LAPNs are both ipsilaterally and contralaterally projecting so this is of interest. Whether or not there are differences in homolateral coupling, this could be discussed in terms of anatomical work of the authors with regards to the populations of LAPNs being silenced and the more caudal LAPNs which are presumably (?) unaffected by the silencing.

These data have been added to figure 6 (previously figure 5) and show that coupling of homolateral limb pairs is unaltered with silencing.

4. Relatedly, information related to efficacy of the technique (i.e. approximate % of the LAPN population affected, distance of spread within lumbar segments when L2 is targeted) would enhance the interpretation as well.

A recent publication from our laboratories (PMID: 33828464) found similar numbers of ipsilateral LAPNs were labeled when using Fluororuby, a 10kD rhodamine dextran amine, (mean number of LAPNs labeled = 135 +/-52) and a dual-viral system (mean number of LAPNs labeled = 126 +/- 46) similar to the dual-viral silencing a silencing system used here. Provided Fluroruby gives an accurate representation of the number of LAPNs, the dual-viral silencing system used here likely silencing >90% of LAPNs. Additionally, eTeNT (produced by the dual-viral silencing system) has extremely high catalytic activity (PMID:7527117) that is active at very low levels. Which also supports that >90% of LAPNs were silenced in the current study. As this information will aid in the interpretation of the current study, it has been added to the “Spared LAPNS express synapse-silencing eTeNT.EGFP chronically” section of the results (lines 140-143).

5. There are several possible explanations for the results offered which are feasible based on the presented data. It seems that these may be possible to narrow down based on tissue from these experiments. Specifically, a quantification of LAPN terminals (or non-somatic fluorescence) in lumbar and cervical cord in these SCI rats compared to that in the uninjured may suggest, first, the degree to which terminals are lost after SCI (or an approximate proportion that are spared) and, second, if there is significant sprouting at either level. Since lumbar terminals were sparse in the intact, are there terminals seen in lumbar cord after SCI? Presence or absence may lend support to either the long loop effects or local gain of aberrant connections.

These are excellent questions/points raised by the Reviewer but are beyond the scope of the current study. To correctly address these questions would require additional cohorts of animals and use of a dual-viral system specifically designed for anatomic quantification. The axons of long propriospinal neurons (spanning > 6 spinal levels) are positioned superficially in the lateral and ventrolateral white matter which increases the likelihood that substantial numbers of these axons remain anatomically intact post-SCI (PMID: 15514981).